# ROGA: Scaling Generalist Agents for Office Productivity Tasks via Tool Generation

**Mugeng Liu**[1]  **Xiaojun Ma**[5,†]  **Yuhang Xie**[3]  **Qin Chen**[4]  **Xuanzhe Liu**[1]  **Yun Ma**[2,†]

[1]School of Computer Science, Peking University  [2]Institute for Artificial Intelligence, Peking University
[3]School of Software and Microelectronics, Peking University
[4]School of Intelligence Science and Technology, Peking University  [5]Individual Researcher
lmg@pku.edu.cn   maxiaojun.pku@gmail.com   yuhangxie@stu.pku.edu.cn
chenqink@pku.edu.cn   xzl@pku.edu.cn   mayun@pku.edu.cn

[†]Corresponding Authors

## Abstract

Automatic tool generation (ATG) has emerged as a key approach to enable the automatic adaptation across diverse tasks within a single generalist agent. Despite their potential, we argue that current ATG agents, often built on reactive paradigms, fail to effectively adapt to realistic environments requiring long-term reasoning and stateful interaction, particularly in office ecosystems. We empirically show that current ATG agents underperform by up to 27.43%. This performance degradation stems from three fundamental limitations of prevailing agent paradigms: (1) a failure to build a coherent world model from long, partially observable contexts; (2) a memory-less execution model where stateless actions fail to track state evolution during iterative tasks; and (3) a static capability generation model focusing on one-shot tool generation for immediate needs, thereby forcing redundant regeneration for similar steps.

To address these fundamental limitations, we propose ROGA, which instantiates a new agent paradigm for long-horizon, stateful environments. ROGA moves beyond simple reactive loops by introducing four foundational algorithmic innovations: (1) **Active World Modeling**, an iterative process where the agent actively probes the environment to construct its own world model; (2) a **Persistent Symbolic Memory** that explicitly tracks the state evolution for temporal reasoning; and (3) a **Dynamic Capability Evolution** model for long-term adaptation and meta-learning on the agent's own capabilities. Comprehensive experiments on widely used benchmarks show that ROGA consistently outperforms existing ATG agents by up to 13.64%. These results underscore ROGA's potential to advance the ATG paradigm, delivering a practical pathway toward building sustainable generalist agents in realistic environments.

## 1 Introduction

Large language model (LLM)-based agents, equipped with tool-using capabilities, have shown considerable promise across a multitude of applications. With the ongoing expansion of applications, encompassing data analytics (Chen et al., 2025), computer operations (Zhang et al., 2024; Sager et al., 2025), and web browsing (OpenAI, 2025a), increasing attention is turning toward *generalist agents*, which can handle diverse tasks within a unified agent (Qiu et al., 2025). To achieve diverse task adaptation, current generalist agents primarily rely on manually crafted, fixed tool sets for each task Lu et al. (2025); Hu et al. (2025). The construction of these tool sets not only demands huge engineering effort but could also fail to cover the specific requirements of open-ended tasks.

To mitigate this issue, the paradigm of automatic tool generation (ATG) has emerged. ATG enables generalist agents to create tools on the fly when existing tools cannot meet the needs of tasks, thereby supporting broader task generalization automatically without extensive human efforts. Recent efforts on ATG have focused on enhancing the quality of generated tools and integrating ATG into generalist agents (ATG agents) to enable automatic adaptation across general tasks. For example, techniques such as *Craft* (Yuan et al., 2024), *Trove* (Wang et al., 2024b), and *Creator* (Qian et al., 2023) employ prompt engineering, code retrieval, and tool refinement to produce reliable

tools. ATG agents like *AutoAgent* (Tang et al., 2025) and *Alita* (Qiu et al., 2025) incorporate ATG mechanisms into generalist agents to support fundamental tasks like mathematical calculations, tool usage, and basic computer operations.

Although ATG agents perform well on simple, stateless tasks, we argue that the *prevailing agent paradigms* are fundamentally limited in real-world utility. This limitation becomes starkly evident when agents face tasks requiring long-term, stateful reasoning within partially observable environments. We select the open-ended office ecosystem, involving Excel, Word, and PowerPoint, as our primary testbed because it is a typical example of this challenging problem class. Moreover, office tasks are not only ubiquitous, supporting workflows that consume billions of user hours daily (Li et al., 2023), but their inherent complexity directly exposes the core flaws of current agent designs.

We identify three inherent drawbacks of current ATG paradigms, which significantly undermine their practical applicability. First, they **fail in building a coherent world model**. Prevailing paradigms rely on passive state perception by accessing the complete picture of the world. However, due to context window limits of LLMs, they cannot capture complete and fine-grained details from lengthy, partially observable office documents. Second, they are constrained by a **memory-less execution model**. The stateless nature of tool calls in existing paradigms breaks the chain of state continuity. They lack an explicit mechanism to track the evolution of shared states across long action chains, a capability essential for complex, iterative tasks such as modifying the same file object across multiple steps. This leads to a loss of context and subsequent errors. Third, they are constrained by **static capability generation**. Prevailing paradigms focus on one-shot tool generation for immediate needs, lacking mechanisms for long-term capability evolution. This leads to redundant regeneration.

To quantitatively highlight the limitations of current ATG agents, we conduct a motivation study comparing three representative generalist agents against a domain-specialized agent in realistic office tasks. The results reveal that current ATG agents consistently underperform compared to the specialized agent, with significant performance degradations of up to 27.43%. This substantial decline underscores the need for further refinement of current ATG agent paradigms. Given their broad practical impact and inherent complexity, office tasks represent a core domain in which any agent aspiring to be a true generalist must be proficient.

To address these fundamental limitations, we introduce ROGA, a novel agent framework that instantiates a new paradigm for structured, state-aware reasoning and adaptation. ROGA is distinguished by three pivotal algorithmic innovations. (1) **Active World Modeling.** Instead of passively perceiving the whole picture of the environment, ROGA actively probes it to build a rich semantic world model. This iterative cognitive process overcomes the limitations of partial observability inherent in long-term environments by adaptively generating specialized comprehension tools to capture world metadata. (2) **Persistent Symbolic Memory.** To counter the memory-less execution model, ROGA employs a persistent symbolic memory. This serves as an explicit, structured working memory that tracks the evolution of the world state across long action chains. It ensures state continuity for iterative operations on shared objects, preventing the context loss inherent in stateless tool calls. (3) **Dynamic Capability Evolution.** To replace the static, reactive generation model, ROGA introduces a mechanism for dynamic capability evolution. This allows the agent to perform meta-learning on its own capability set, enabling it to retain, refine, and reuse its skills over time. This approach eliminates redundant regeneration and fosters long-term adaptation.

We comprehensively evaluate ROGA against widely-used benchmarks involving automatic reasoning and manipulation in open-ended office ecosystems, including Excel, Word, and PowerPoint. Results demonstrate that ROGA consistently outperforms current ATG agents across comprehensive office tasks by up to 13.64% task success rate. Notably, in challenging table-based tasks, ROGA matches and even exceeds the performance of agents tailored for table tasks. This highlights the huge potential of the ATG paradigm in handling real-world tasks requiring long-term, stateful reasoning. Additionally, experiments on mathematical datasets demonstrate that ROGA also maintains superior generalization performance in non-office tasks. These findings underscore ROGA's potential as a viable pathway toward sustainable ATG agents in open-ended, realistic environments.

In summary, this paper makes the following key contributions[1].

---

[1]The source code of ROGA is publicly available at `https://github.com/morgen52/roga`.

- We identify that prevailing agent paradigms are fundamentally ill-suited for long-term, stateful tasks. We highlight three core algorithmic flaws: the inability to build world models from partially observable contexts, the loss of state continuity from memory-less execution, and the failure of long-term capability evolution due to static capability generation.
- We propose a new agent paradigm, instantiated in ROGA, that overcomes these flaws through three foundational innovations: Active World Modeling, Persistent Symbolic Memory, and Dynamic Capability Evolution.
- Comprehensive experiments demonstrate that ROGA establishes a new state-of-the-art. It significantly outperforms existing ATG agents and even specialized agents, demonstrating the effectiveness of our proposed paradigm in long-term, stateful environments.

## 2 BACKGROUND

### 2.1 RELATED WORK

**Tool Retrieval** advances tool selection to enhance LLM-based agents. Toolformer (Schick et al., 2023) teaches models to self-learn when and how to call APIs from unlabeled data. ToolLLM (Qin et al., 2024) fine-tunes LLMs on large synthetic datasets for accurate multi-tool invocation. Tool-Gen (Wang et al., 2025a) embeds tools as virtual tokens to unify retrieval and generation. Although these methods improve retrieval and invocation, they rely on predefined toolsets, limiting adaptability to tasks beyond existing tools in open environments.

**Automatic Tool Generation** (Cai et al., 2024) creates tools from tool specifications. Advanced methods incorporates retrieval to generate reliable tools. Code retrieval (Wölflein et al., 2025) reuses existing snippets with self-correction, while prompt-based retrieval (Yuan et al., 2024; Tan et al., 2024) leverage engineered prompts for richer generation context. Other techniques (Qian et al., 2023; Wang et al., 2024b) refine generated tools through execution-based testing. However, all the methods require well-defined tool specifications, limiting their applicability to open-ended office tasks where specifications need to be inferred through nuanced reasoning over diverse file contexts.

**ATG Agents** (Wang et al., 2025b) mark a shift toward generalist agents that dynamically generate task-specific tools, moving beyond fixed tool sets. Recent advances (Tang et al., 2025; Qiu et al., 2025) adopt frameworks where agents autonomously search and create modular executable code as tools for task completion. However, current solutions remain limited to basic tasks (e.g., mathematics and basic computer usage) and fall short in long-term, stateful office environments requiring rich file context and complex tool orchestration.

**Office-Specific Agents** have attracted attention for automating labor-intensive workflows and delivering high commercial value (Microsoft, 2025; Google, 2025). Recent work introduces challenging benchmarks from real-world office forums (Li et al., 2023; Wu et al., 2025) and designs specific pipelines for individual applications such as Excel (Li et al., 2023; Chen et al., 2025). However, these dedicated agents often depend on predefined tools and workflows, limiting their generalization across diverse office tasks and environments.

Table 1: Performance comparison of three ATG agents and a specialized agent. The best results are marked with *.

| Method | TableBench | | SheetCopilotBench | |
|---|---|---|---|---|
| | Exec@1 ↑ | Pass@1 ↑ | Exec@1 ↑ | Pass@1 ↑ |
| AutoAgent (GPT-4.1) | 51.58 | 24.04 | 58.37 | 14.48 |
| OctoTools (GPT-4.1) | 51.81 | 25.73 | 30.32 | 14.48 |
| OWL (GPT-4.1) | 81.94 | 46.28 | 85.52 | 19.91 |
| SheetAgent (GPT-4.1) | 89.39* | 51.47* | 98.64* | 23.53* |

### 2.2 MOTIVATION STUDY

We conduct a motivation study to quantitatively explore the limitations of current ATG agents.

**Experiment Setup.** Our empirical evaluation is centered on spreadsheet tasks. This focus is driven by three unique factors that are not present in other office tasks: their role as a representative and ubiquitous office automation scenario, the availability of established, real-world spreadsheet benchmarks, and the existence of a specialized agent explicitly designed for this domain.

Specifically, we compare three generalist agents (Octotools (Lu et al., 2025), OWL (Hu et al., 2025), AutoAgent (Tang et al., 2025)) with a spreadsheet-specific agent (SheetAgent (Chen et al., 2025)). We evaluate the agents on two widely used spreadsheet benchmarks: TableBench (Wu et al., 2025) for table question answering, and SheetCopilotBench (Zhu et al., 2025) for table manipulation. We use the GPT-4.1 as the backbone LLM for all the agents. Additionally, we adopt Exec@1 to quantify the percentage of solutions that execute without errors, and Pass@1 to assess the percentage of successful task completion.

**Results Analysis.** As shown in Table 1, ATG agents exhibit significantly lower performance than the domain-specific SheetAgent across benchmarks and metrics. Notably, SheetAgent surpasses AutoAgent, one of the state-of-the-art (SOTA) ATG agents, by a large margin regarding Pass@1, with a performance gap of up to 27.43% on TableBench. These substantial performance gaps underscore the critical limitations of prevailing ATG agent paradigms, highlighting their failure to generalize to crucial real-world tasks, particularly widely used office tasks.

## 3  ROGA DESIGN

In this section, we provide an overview and introduce the three core innovations of ROGA that address the identified limitations.

### 3.1  OVERVIEW

ROGA overcomes the inherent limitations of current ATG agents by introducing a comprehensive framework designed to handle long-term, stateful office tasks effectively. As illustrated in Figure 1, ROGA is composed of the following components:

- **Planner**: Orchestrates the agent's decision flow based on the Active World Model.
- **Executor**: Provides Persistent Symbolic Memory for long-term, stateful action.
- **Tool Manager**: Enables Dynamic Capability Evolution, along with the full lifecycle management of tools.
- **Tool Generator and Validator**: Generates and refines tools as part of the Dynamic Capability Evolution process.

**Definition.** The workflow of ROGA is modeled as a discrete-time decision process. At each step $t$,

$$\mathcal{S}_t = (F_t, M_t, \mathcal{T}_t)$$
$$a_t \sim \pi(\mathcal{S}_t) \quad \text{where} \quad a_t \in \{\text{TOOL GENERATION}, \text{TOOL EXECUTION}, \text{DONE}\}$$
$$\mathcal{S}_{t+1} = \delta(\mathcal{S}_t, a_t)$$

where $S_t$ is the agent state, $F_t$ the long, partially observable file contexts that need comprehension and operation, $M_t$ the Persistent Symbolic Memory containing current code context and semantic comprehension, and $\mathcal{T}_t$ the self-evolved capability set. The decision function, denoted as $\pi$, selects an action from the predefined action space based on the current state of the agent. Subsequently, the state transition function, represented by $\delta$, applies the selected action through the relevant module to update the agent's state.

### 3.2  ACTIVE WORLD MODELING FOR PARTIALLY OBSERVABLE CONTEXT

To overcome the limitation of passive perception in current agent paradigms, which assume full observability of input context, ROGA introduces **Active World Modeling (AWM)**. This paradigm addresses the core challenge of LLM agents: understanding environments that are too large to fit into a context window. The innovation of AWM is its active and iterative nature. Instead of a single, passive "read" step in the current paradigm, ROGA enters a meta-cognitive loop where it learns to iteratively probe and understand the environment by generating specialized comprehension tools. This is instantiated through a dedicated comprehension phase before operational steps in ROGA.

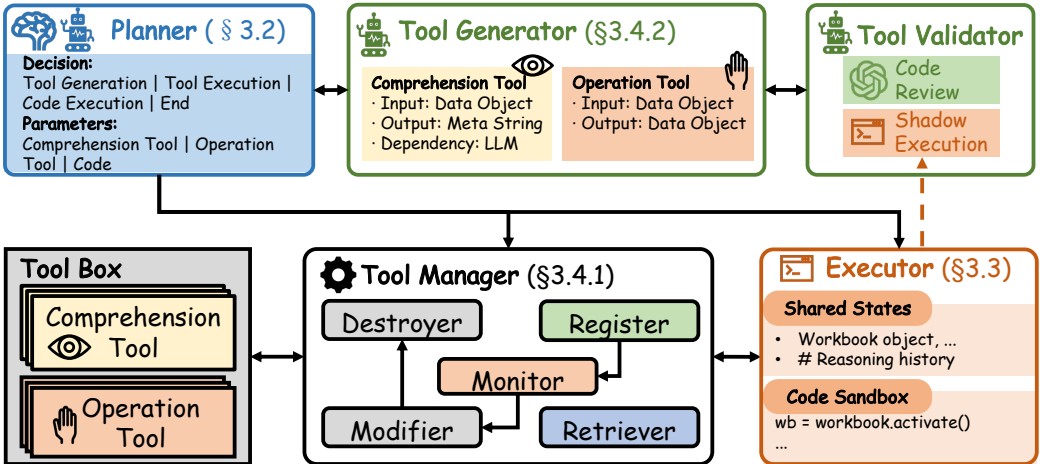

Figure 1: Overview of ROGA.

Specifically, when the planner $\pi$ detects that its world model (stored in persistent symbolic memory $M_t$) is insufficient for the task, it initiates the AWM process. Instead of passively attempting to ingest the entire, often overwhelmingly large, file context, ROGA takes an active, iterative, meta-cognitive loop. The agent identifies the knowledge gap, formulates a specific information-seeking sub-goal (e.g., "What are the names of all worksheets containing a pivot table?"), and then dynamically generates and invokes specialized comprehension tools to precisely answer the question.

This process is a form of targeted environmental probing, which repeats iteratively: ROGA continuously identifies knowledge gaps and generates new tools to read, retrieve, and map out the world model piece by piece. The outcome (i.e., the answer to the information-seeking sub-goal) is then integrated into the world model in $M_t$, effectively filling the gap in its context comprehension. These dynamically generated tools, which can extract specific details like Excel formulas, Word styles, or PowerPoint layouts, are added to the agent's capability set $\mathcal{T}_t$ for future reuse. By meticulously processing these details, ROGA minimizes the risk of oversight and misinterpretation of file contexts, thereby setting a robust basis for accurate and context-aware operations.

After the comprehension phase, the operation phase utilizes insights from the world model in $M_t$ to execute tasks with enhanced precision. This phase involves applying operation-specific tools, generated from the refined context understanding. In this context, operations are guided by comprehensive context analysis, aligning with user objectives and the complexities of the file content.

By grounding operations in a thorough world model built via the active comprehension phase, ROGA effectively addresses the limitations of passive perception and missing embedded details. This AWM approach establishes a robust basis for subsequent actions, significantly enhancing reliability in long-term, partially observable office environments. Additionally, the clear separation of comprehension and operation tools facilitates modular tool design and generation, promoting tool reuse in complex, open-ended task settings.

## 3.3 PERSISTENT SYMBOLIC MEMORY FOR STATEFUL ACTION

Prevailing agent paradigms are constrained by a memory-less execution model, where tool invocations are treated as isolated, stateless function calls. This fundamentally breaks the chain of causality required for complex, iterative tasks. To counter this, ROGA introduces **Persistent Symbolic Memory (PSM)**, an innovative mechanism that endows the agent with a structured, evolving representation to track the world state across time.

At its core, PSM is not merely a shared memory space, but an explicit symbolic state ledger ($M_t$). It maintains symbolic handles to all intermediate artifacts (e.g., code objects, file handles, or dataframes) and tracks their state evolution throughout the task. This allows a tool in step $t$ to directly reference and operate on an object created in prior steps $t-i, i \in [1, t-1]$, enabling temporal reasoning and stateful interaction, a capability absent in stateless models.

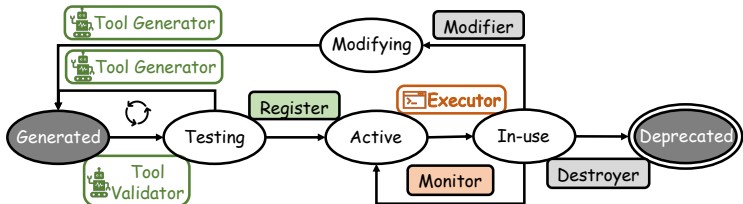

Figure 2: Finite-state machine of the tool lifecycle management.

ROGA realizes this mechanism through an innovative state-sharing sandbox that enforces the principles of PSM. The sandbox provides a unified record and sharing API for the agent to save and obtain intermediate states, which facilitates the uniform sharing of the symbolic state ledger $M_t$ among all tools. This is is crucial for supporting iterative updates on shared objects (e.g., $F_t$).

Furthermore, to guarantee the integrity of this temporal state chain, the PSM framework mandates two key algorithmic properties, implemented by the sandbox: **atomicity** and **reversibility**. All tool invocations are executed atomically: An operation either fully succeeds and commits its changes to create state $M_{t+1}$, or it fails and the system entirely rolls back to the previous consistent state $M_t$. This state-rollback mechanism, which records and can replay the history of state-modifying operations, ensures that execution errors do not corrupt the state chain.

By formalizing state management through the Persistent Symbolic Memory paradigm, ROGA transforms a sequence of independent actions into a coherent, stateful workflow. This enables safe and seamless tool coordination over long horizons, boosting the agent's reliability for long-term, stateful office tasks.

## 3.4 DYNAMIC CAPABILITY EVOLUTION

To move beyond the static, one-shot capability generation that leads to redundant effort and fails to learn from past experience, ROGA introduces Dynamic Capability Evolution. This mechanism represents a paradigm shift, treating tool generation not as a series of isolated events, but as a continuous meta-learning process on the agent's own capability set ($\mathcal{T}_t$). It enables the agent to map its exist capabilities to current context spaces. This paradigm is realized through two tightly integrated algorithmic components: a formal framework for managing the lifecycle of capabilities to retain, refine, and reuse them over time; and a situated self-correction process that acts as the core engine of evolution.

### 3.4.1 A FORMALISM FOR CAPABILITY LIFECYCLE

Instead of an ad-hoc collection of tools and repeatedly regenerating similar tools, ROGA models its capability set $\mathcal{T}_t$ using a computational formalism based on a Finite-State Machine (FSM), as shown in Figure 2. This FSM provides a formal structure for the meta-learning process, defining the evolutionary stages of any given capability.

Each state represents a distinct phase in a capability's life-cycle:

- **Generated**: A newly created capability template, potential mapping to the task space, formulated as a candidate hypothesis, awaiting validation.
- **Active**: A validated and reliable capability that has validated its correct mapping in current tasl context, now part of the agent's reusable skill repertoire.
- **In-use**: A transient state indicating the capability's active mapping onto the current task context space, indicating it is being executed to address a specific, immediate goal.
- **Modifying**: A state triggered when the capability fails to map successfully onto the current task context space. The capability undergoes targeted refinement driven by feedback from this specific failure context to improve its situational fitness.
- **Deprecated**: A terminal state assigned to a capability after a history of repeated mapping failures across different task contexts. It is permanently removed from the active capability set to prevent propagating errors, reflecting a conclusion that this capability template lacks general utility.

By formalizing the capability lifecycle, this FSM transforms the agent's behavior from purely reactive generation to a proactive, structured learning process to iteratively refine the generated capability templates in evolving task context spaces. The success or failure of each capability-to-task mapping attempt, captured while a capability is in the 'In-use' state, serves as an empirical fitness score. This score can guide the selection probability during retrieval and the evolutionary path of each capability. This structure is the backbone that enables the agent to learn from experience and facilitate systematic capability evolution.

### 3.4.2 SITUATED SELF-CORRECTION

The transitions between states in the lifecycle formalism are driven by Situated Self-Correction (SSC), a novel validation architecture that serves as the core engine for capability evolution. The core innovation of SSC is its *situated* nature: the capability correction is not a generic, offline process but is deeply grounded in the agent's specific, dynamic state ($M_t$) and immediate goals at the moment of evaluation. This contrasts sharply with traditional methods that validate code in a decontextualized vacuum environment, which is insufficient for long-term, stateful tasks.

The SSC mechanism operates as a tight, iterative loop, beginning with the agent proposing a new or modified tool, i.e., a candidate capability for action. This candidate is then subjected to a rigorous, situated validation process within the precise context of the current state $M_t$, unfolding across two complementary channels. The channels include state-aware functional testing and semantic intent validation. For functional testing, the capability is performed in a shadow sandbox that is a perfect replica of the Persistent Symbolic Memory $M_t$. This moves beyond syntactic correctness to answer a state-aware question: does the capability execute successfully and produce the expected effect in the current agent states? This provides direct, state-aware functional validation. Concurrently, the semantic validation phase involves an in-depth examination of the tool's alignment with the semantic nuances of the task context (including user intent and file context). This phase leverages LLMs for code review to ensure that the capabilities are contextually appropriate, capable of comprehending or operating on the current file $F_t$, and correctly using parameters within the current code context of $M_t$. This process mitigates functional issues associated with subtle semantic errors.

The combined feedback from this dual-channel validation guides the capability evolution. A successful outcome promotes a 'Generated' or 'Modifying' capability to 'Active', whereas a failure provides the precise error context to trigger a transition to 'Modifying', thereby driving the meta-learning loop to generate more refined capability candidates.

Driven by continuous, context-aware validation, the SSC iteratively evolves capabilities across diverse contexts. This situated self-correction is cornerstone of the capability evolution process, ensuring the robust and efficient capability evolution in long-term, stateful tasks.

## 4 EXPERIMENTS

### 4.1 EXPERIMENTAL SETUP

**Benchmark.** We extracted all the 42 and 110 tasks related to Excel, Word, and PowerPoint from the comprehensive benchmarks WindowsAgentArena (WAA) (Bonatti et al., 2025) and OSWorld (Xie et al., 2024) to evaluate ROGA's capabilities across diverse office types. To compare the performance of ROGA with domain-specific agents in the specific office domain, we conducted experiments in the spreadsheet domain using TableBench (886 tasks) (Wu et al., 2025) and SheetCopilotBench (SCB) (221 tasks) (Li et al., 2023). Additionally, to examine whether ROGA 's design can sustain the reasoning capabilities achieved by generalist agents in established domains, we evaluated ATG agents on 100 math problems randomly selected from Math500 (Lightman et al., 2024) and 100 problems randomly selected from a challenging multi-task understanding benchmark, MMLU-Pro (Wang et al., 2024a). A detailed description of these benchmarks and the rationale for their selection is provided in Appendix B.

**Baselines.** We compare ROGA against three representative ATG agents, including AutoAgent (Tang et al., 2025), Octotools (Lu et al., 2025), and OWL (Hu et al., 2025). These agents exhibit state-of-the-art performance on general tasks with few reasoning steps and show considerable diversity in their designs. Note that OWL is a multi-agent system that assigns tasks to predefined, domain-

Table 2: End-to-end execution success rate (Exec@1, %) and task success rates (Pass@1, %) on office tasks. The best results are marked with *, and the second-best are underlined.

| Method | OSWorld | | WAA | | GAIA-Office | |
|---|---|---|---|---|---|---|
| | Exec@1 ↑ | Pass@1 ↑ | Exec@1 ↑ | Pass@1 ↑ | Exec@1 ↑ | Pass@1 ↑ |
| AutoAgent (GPT-4.1) | 62.73 | 17.27 | 73.81 | 23.81 | 27.59 | 13.79 |
| OctoTools (GPT-4.1) | 63.64 | 18.18 | 69.04 | 26.19 | 42.31 | 11.54 |
| OWL (GPT-4.1) | 79.09 | 16.36 | 76.19 | 19.05 | 92.31 | 73.08 |
| ROGA (GPT-4.1) | 95.45* | 31.82* | 97.62* | 28.57* | 96.15* | 76.92* |
| *Different Backbone LLMs* | | | | | | |
| ROGA (OpenAI-o3) | 31.82 | 10.00 | 57.14 | 19.05 | 19.23 | 15.38 |
| ROGA (Claude Sonnet 4) | 92.73 | 29.09 | 92.86 | 26.19 | 80.77 | 53.85 |
| *Ablation Study* | | | | | | |
| ROGA (GPT-4.1) | | | | | | |
| - w/o Active World Modeling | 96.36 | 20.00 | 95.24 | 19.05 | 76.92 | 50.00 |
| - w/o Situated Self-Correction | 95.45 | 23.64 | 92.86 | 14.29 | 80.77 | 46.15 |
| - w/o Persistent Symbolic Memory | 89.09 | 19.09 | 85.71 | 21.42 | 80.77 | 46.15 |
| - w/o Capability Lifecycle | 90.00 | 29.09 | 88.10 | 26.19 | 76.92 | 53.85 |

specialized agents. For the office tasks in our benchmarks, it utilizes dedicated agents like a *ExcelAgent* for spreadsheet tasks and a *DocumentAgent* for processing Word and PowerPoint files. Thus, OWL serves as a domain-specific baseline for the office tasks. Additionally, we employ an advanced spreadsheet-specific agent, SheetAgent (Chen et al., 2025), to benchmark performance on more challenging spreadsheet tasks. More detailed descriptions of these benchmarks and the rationale for their selection can be found in Appendix B.

## 4.2 OVERALL PERFORMANCE

Table 2 presents a comprehensive evaluation of ROGA against state-of-the-art (SOTA) baselines across three prominent office automation benchmarks, including OSWorld, WindowsAgentArena, and GAIA-Office. The results demonstrate that ROGA achieves superior performance, establishing new SOTA results on all benchmarks for both metrics. This consistent dominance highlights the robustness and generalizability of ROGA in office tasks requiring long-step reasoning.

Specifically, on the OSWorld benchmark, ROGA surpasses the best baselines by 16.36% in Exec@1 and 13.64% in Pass@1. This trend is also evident on the WAA benchmark, where ROGA outperforms the nearest baselines by 21.43% in Exec@1 and 2.38% in Pass@1. Moreover, ROGA exhibits notable reasoning capabilities in office QA tasks from the GAIA-Office benchmark, with over 95% Exec@1 and over 75% Pass@1. These results collectively underscore the advanced capabilities of ROGA in understanding length file context, generating and executing reliable tools, and correctly handling the tool interactions in open-ended office environments with long reasoning steps.

**Impact of Different Backbone LLMs.** To evaluate the sensitivity of ROGA to the choice of backbone LLM, we conduct experiments using three distinct LLMs, including GPT-4.1 (OpenAI, 2025b), OpenAI-o3 (OpenAI, 2025c), and Claude Sonnet 4 (Anthropic, 2025). As shown in Table 2, ROGA with GPT-4.1 achieves exceptional performance across benchmarks. Meanwhile, ROGA with Claude Sonnet 4 delivers competitive yet slightly lower results. Notably, ROGA with OpenAI-o3 experiences a substantial performance decline, underscoring its inadequacy for complex office automation tasks. This decline is primarily attributed to the smaller context window of OpenAI-o3 compared to GPT-4.1 and Claude Sonnet 4, which limits its capacity to handle lengthy file contexts and extended reasoning steps required in office tasks. These findings underscore the significant challenges that agents encounter in accurately processing and comprehending extensive file contexts in office environments. Based on these results, we employ the best-performing model, GPT-4.1, as the LLM backbone for all baselines in our experiments.

**Ablation Study.** Table 2 presents the results of the ablation study, which systematically evaluates the contribution of each core component in ROGA by removing them individually.

• Removing Active World Modeling results in a significant decline in Pass@1 across all benchmarks, while Exec@1 remains largely unaffected. This indicates without its active probing mechanism, the

'

Table 3: Token usage, cost, and reasoning steps comparison. Agents are equipped with GPT-4.1. Costs are calculated based on OpenAI pricing ($3/1M input tokens, $12/1M output tokens).

| Method | OSWorld | | | | WinArena | | | | GAIA-Office | | | |
|---|---|---|---|---|---|---|---|---|---|---|---|---|
| | In(k) | Out(k) | Cost($) | Step | In(k) | Out(k) | Cost($) | Step | In(k) | Out(k) | Cost($) | Step |
| ROGA | 56.10 | 5.77 | 0.26 | 13.56 | 64.96 | 3.27 | 0.23 | 10.78 | 46.66 | 5.52 | 0.21 | 12.67 |
| AutoAgent | 15.35 | 1.17 | 0.06 | 8.56 | 14.14 | 1.09 | 0.06 | 7.50 | 37.53 | 2.72 | 0.15 | 18.75 |
| OctoTools | 19.43 | 4.05 | 0.11 | 9.17 | 12.72 | 3.20 | 0.08 | 7.33 | 31.78 | 5.43 | 0.16 | 12.78 |
| OWL | 37.22 | 1.21 | 0.13 | 12.00 | 39.28 | 1.31 | 0.14 | 12.61 | 27.60 | 0.95 | 0.09 | 9.56 |

Table 4: Performance on challenging spreadsheet tasks, using GPT-4.1 as backbone LLM. Results with * indicate the best.

| Method | TableBench | | SheetCopilotBench | |
|---|---|---|---|---|
| | Exec@1 ↑ | Pass@1 ↑ | Exec@1 ↑ | Pass@1 ↑ |
| SheetAgent | 89.39 | 51.47 | 98.64 | 23.53 |
| ROGA | 95.03* | 56.09* | 100.00* | 25.91* |

agent fails to build a coherent world model from the partially observable file, leading to incorrect outcomes even if individual actions can be executed. This highlights the framework's core strength in handling partial observability.

• Removing the Situated Self-Correction during tool generation causes declines in both Exec@1 and Pass@1. This suggests that without the state-aware validation process, generated tools are more prone to execution failures or semantic inconsistencies, demonstrating the importance of grounding capability evolution in the current task context.

• Removing the Persistent Symbolic Memory produces the most substantial reductions in both metrics. This underscores its criticality for maintaining state continuity in iterative tasks. Without it, the agent loses track of changes to shared objects across steps, leading to cascading failures characteristic of long-term, stateful problems.

• Removing Capability Lifecycle Management impairs performance in GAIA-Office. In OSWorld and WAA, the Pass@1 rate shows a slight decline, likely due to the dual-reflection mechanism enhancing the accuracy of repeat generation. Further analysis reveals that the absence of tool reuse leads to a $1.54\times$ increase in average reasoning steps, indicating higher reasoning overhead in tool generation. This indicates that without a mechanism for capability reuse and refinement, the reliance on redundant, reactive generation increases both execution errors and reasoning overhead, a key challenge in tasks with recurring sub-problems.

A more in-depth analysis of the primary failure modes corresponding to each ablated component, along with a granular performance breakdown by file type, is provided in Appendix C.

**Cost Analysis.** As shown in Table 3, ROGA's higher token usage is not a flaw but a deliberate investment in a superior reasoning paradigm that leads to a higher success rate. Note that baseline agents often suffer from "high-cost failures", wasting their budgets on complex but flawed plans based on flawed assumptions about the partially observable world. In contrast, ROGA's upfront cost is strategically allocated to actively probe the environment to build a verified world model via *Active World Modeling (AWM)*, thereby de-risking the entire task. Our further analysis shows that this mechanism takes approximately 40% of the cost. Another 20% comes from the rigorous validation within *Dynamic Capability Evolution (DCE)*. Crucially, the DCE framework itself serves as a built-in cost mitigation strategy, operating on an "invest once, use many times" principle that amortizes the initial generation cost over time by reusing validated capabilities. This is empirically validated by our ablation study, which shows that removing the capability lifecycle management resulted in a $1.54\times$ increase in reasoning steps due to redundant tool regeneration. Therefore, ROGA's cost structure reflects a strategic trade-off: a higher initial investment in robustness and correctness that prevents costly failures and becomes more efficient over time.

## 4.3 PERFORMANCE ON CHALLENGING SPREADSHEET TASKS

In the motivation study, ATG agents consistently underperformed compared to the advanced table-specific agent, SheetAgent, on challenging real-world spreadsheet tasks. To explore whether gener-

Table 5: Performance on non-office tasks, using GPT-4.1 as the backbone LLM.

| Method | Math500 | | MMLU-Pro | |
|---|---|---|---|---|
| | Exec@1 ↑ | Pass@1 ↑ | Exec@1 ↑ | Pass@1 ↑ |
| AutoAgent | 84.00 | 54.00 | 100.00 | 78.00 |
| OctoTools | 100.00 | 70.00 | 97.00 | 64.00 |
| OWL | 99.00 | 71.00 | 93.00 | 65.00 |
| ROGA | 99.00 | 70.00 | 100.00 | 76.00 |

alist agents employing ATG paradigm can outperform specialized agents in a specific domain, we also evaluate ROGA using TableBench and SheetCopilotBench.

Results (Table 4) demonstrate that ROGA outperforms the domain-specific agent on both benchmarks in both metrics. This demonstrates that ROGA not only excels in general office tasks across file types but also achieves SOTA performance in specific domains, highlighting its versatility and advanced reasoning capabilities. Moreover, this finding underscores the promising potential of the ATG agent for adapting to a broader range of realistic, open-ended environments. By addressing the inherent shortcomings of current ATG agent paradigms, ATG agents can match or even surpass the performance of specially designed agents.

### 4.4 GENERALIZATION ON NON-OFFICE TASKS

As shown in Table 5, ROGA achieves strong performance on par with the best baselines. This result is crucial as it confirms the generality of our paradigm. The performance difference between office and non-office tasks is not because ROGA is tailored for office applications, but because ROGA is designed to solve a specific category of problems that current ATG agents cannot handle: long-term, stateful tasks in partially observable environments.

Office tasks are well defined by these characteristics, which activate ROGA's core innovations. **Active World Modeling** is critical for understanding large, partially observable files. **Persistent Symbolic Memory** is essential for tracking state across iterative, dependent, stateful actions. **Dynamic Capability Evolution** provides value in long-term tasks with recurring sub-problems. In contrast, tasks like mathematics are typically stateless and fully observable, where the problem fits in a single context window and can be solved with stateless tool calls. In such cases, ROGA's advanced mechanisms are not required and thus provide no significant advantage.

Therefore, we draw two key conclusions from these results. First, ROGA's significant gains on office tasks validate the effectiveness of our paradigm for this challenging problem class. Second, its comparable performance on math (Math500) and knowledge-intensive tasks (MMLU-Pro) confirms its generality, demonstrating that ROGA extends agent capabilities to a new, challenging domain without sacrificing performance on established problems where those new capabilities are not needed.

## 5 CONCLUSION

To our best knowledge, we are the first to identify the critical limitations of prevailing ATG agent paradigms in long-term, stateful environments: their failure to build coherent world models, memory-less execution, and static capability generation. To address them, we propose ROGA, a novel generalist agent that instantiates a new paradigm for long-term, stateful tasks, especially office tasks. ROGA introduces three key innovations: Active World Modeling for deep environmental comprehension, a Persistent Symbolic Memory for stateful, continuous actions, and Dynamic Capability Evolution for long-term adaptation and skill refinement. Extensive experiments show that ROGA significantly outperforms existing ATG agents and even specialized office agents, while maintaining strong generalization in non-office domains. These results highlight ROGA's potential to scale the capability of ATG agents to complex open-ended scenarios in the real world, paving the way for more adaptive and robust generalist agents.

## ACKNOWLEDGMENTS

This work was supported by the National Natural Science Foundation of China under the grant numbers 62595734 and 62325201, the Key Laboratory of High Confidence Software Technologies (Peking University), the Ministry of Education, and the Center for Data Space Technology and System, Peking University.

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

## A    STATEMENTS ON LLM USAGE

LLMs were employed to assist in the writing of this paper, primarily for sentence polishing and grammar correction. We have thoroughly reviewed and modified all content generated by LLMs to ensure no inappropriate descriptions.

## B    DETAILED EXPERIMENT SETTINGS

This section introduces the detailed settings of experiments.

### B.1    BENCHMARK SELECTION

• **WindowsAgentArena (WAA)** (Bonatti et al., 2025) consists of automatic manipulation tasks across 15 commonly used Windows applications. We extract all the tasks involving office productivity tools from WAA, obtaining 42 office tasks associated with Excel, Word, and PowerPoint for our evaluation. This benchmark is used to assess the manipulation capabilities of agents across diverse file types.
• **OSWorld** (Xie et al., 2024) is a comprehensive benchmark containing manipulation tasks related to office applications, browser interactions, and file-system operations. For our evaluation, we specifically focus on the office-related tasks, extracting a total of 110 office tasks that cover Excel, Word, and PowerPoint. Each tasks are equipped with handcrafted verification scripts for reliable correctness checking. Compared to WAA, this benchmark contains more challenging manipulation tasks across heterogeneous file types.
• **GAIA-Office** is derived from GAIA (Mialon et al., 2023), a widely utilized benchmark designed for evaluating general-purpose AI assistants. By filtering all office-related tasks, we have constructed GAIA-Office with 26 task cases. These tasks focus on question-answering (QA) within the contexts of Excel, Word, and PowerPoint, aiming to assess the agent's reasoning capabilities across various document types.
• **TableBench** (Wu et al., 2025) is a table question answering benchmark comprising 18 challenging task categories across 886 samples. It is specifically designed to assess critical reasoning capabilities within the domain of spreadsheets, a particularly important category of office tasks. This benchmark evaluates ROGA's ability to perform complex and long reasoning on tabular data. Compared to general benchmarks that emphasize task diversity, this domain-specific benchmark presents greater challenges in terms of reasoning difficulty.
• **SheetCopilotBench (SCB)** (Li et al., 2023) is a spreadsheet manipulation benchmark derived from real-world applications. It comprises 221 tasks that require both comprehension and operational skills in handling tabular data. This benchmark assesses the ability of agents to execute realistic sequences of spreadsheet operations. Although it focuses only on spreadsheets, its manipulation difficulty within this domain is significantly higher than that of general benchmarks involving various file types.
• **Math500** is a widely used mathematical reasoning benchmark derived from the MATH dataset released by OpenAI Lightman et al. (2024). It consists of a carefully curated subset of math problems designed to test a wide range of mathematical reasoning skills. For our study, we randomly selected 100 problems from this benchmark due to resource constraints. This dataset aims to verify whether ROGA's design can sustain the reasoning capabilities that a generalist agent has already achieved in existing domains. Following prior work (Tang et al., 2025), we assessed the agent's mathematical reasoning abilities.

### B.2    BASELINE SELECTION

We compare ROGA against four representative agents, exhibiting a high variety in their design.

• **AutoAgent** (Tang et al., 2025) is a zero-code platform that facilitates the creation, customization, and deployment of agents powered by LLMs. It is an ATG agent designed to generalize across diverse tasks through automatic tool generation, functioning without predefined tools. For office tasks, AutoAgent employs three core agents, Web Search, File System, and Code Agent, for task solving.
• **Octotools** (Lu et al., 2025) is a recent generalist agent designed to streamline multi-tool workflows in complex computational tasks. It offers more than 10 standardized tool cards that cover a wide

range of functionalities, such as code generation and execution. These tool cards facilitate the agent to handle tasks across multiple domains. For office tasks, this baseline relies on a predefined code generation and execution tool.

• **OWL** (Hu et al., 2025) offers an agent system that decomposes tasks into specialized sub-tasks, each managed by a predefined agent type, such as UserAgents, AssistantAgents, ToolAgents, and CodeAgents. This system enables the automation of complex real-world tasks through dynamic collaboration among multiple agents. For office tasks, OWL features predefined, dedicated agent types, including ExcelAgent for spreadsheet tasks and DocumentAgent for processing Word and PowerPoint files.

• **SheetAgent** (Chen et al., 2025) is a domain-specific agent tailored for spreadsheet tasks. It includes a planner dedicated to table analysis and decision-making, an informer that employs SQL-based queries for data retrieval from spreadsheets, and a code executor that runs Python code using the OpenPyXL library within a single Python environment. This design enhances the agent's reasoning and operational capabilities for complex spreadsheet tasks. However, due to the specialized design for spreadsheets, it cannot be generalized to other office tasks or a broader range of applications.

## C IN-DEPTH ANALYSIS

### C.1 FAILURE CASE ANALYSIS

To provide deeper insight into why the components of ROGA are critical, we conducted a systematic analysis of the primary failure modes in long-term, stateful tasks. Our findings, strongly corroborated by the ablation study on OSWorld (Table 2), present the error types below, ranked from most to least critical.

• **Stateful Execution Failures.** This is the most critical error class. The largest performance drop in our ablation study (-12.73% in Pass@1) occurred when removing **Persistent Symbolic Memory (PSM)**. This quantitatively demonstrates that the inability to maintain state across multiple, dependent actions is the primary failure mode. Prevailing agents, operating on a stateless model, frequently fail on iterative tasks (e.g., modifying a data table in step 2 and then creating a chart from that modified table in step 5), as their isolated tool calls break the chain of causality and lose the evolved state of the world.

• **Comprehension Failures.** This is the second-largest source of error. Removing **Active World Modeling (AWM)** caused an 11.82% drop in Pass@1. This shows that failing to build an accurate model of a partially observable environment is another major failure mode. When an agent cannot see the entire context (e.g., a large spreadsheet), it acts on incomplete or incorrect assumptions, leading to fundamental flaws in planning and tool specification, ultimately producing wrong results.

• **Tool Generation Failures.** This is the third most frequent error. Removing **Situated Self-Correction (SSC)** led to a significant 8.18% drop in Pass@1. This indicates that generating faulty tools is a common problem in current paradigms. Without a state-aware validation process, the likelihood of generating syntactically incorrect or semantically misaligned tools increases substantially, leading to incorrect results.

### C.2 LIMITATIONS OF SITUATED SELF-CORRECTION (SSC)

While our Situated Self-Correction (SSC) mechanism is highly effective at catching functional errors and contextual inconsistencies, it is not infallible. Its primary strength lies in validating a tool's correctness against the agent's current state, but it can fail to catch more subtle, higher-level logical errors. To better understand these boundaries, we manually examined 20 cases where SSC was triggered, but the task ultimately failed. Our analysis reveals that the failures typically fall into the following two main categories.

• **Nuanced Semantic Flaws.** This is the most common failure mode. It occurs when a generated tool is functionally correct but performs the wrong semantic action due to ambiguity in the user's request. For example, given a request to "create a chart based on the data in this table," the agent might correctly generate a tool to create the chart and place it on a new sheet. While this action is valid and executes successfully, the task may fail if the benchmark's ground truth has an unstated

Table 6: Performance breakdown by file type on the OSWorld benchmark. ROGA's consistent high performance contrasts with the erratic results of baseline agents.

| File Type | Agent | Exec@1 (%) | Pass@1 (%) |
|---|---|---|---|
| **Word** | **ROGA** | **100.00** | **42.86** |
| | AutoAgent | 66.67 | 28.57 |
| | OctoTools | 80.95 | 14.29 |
| | OWL | 90.48 | 4.76 |
| **Excel** | **ROGA** | **91.49** | **31.91** |
| | AutoAgent | 51.06 | 10.64 |
| | OctoTools | 46.81 | 14.89 |
| | OWL | 72.34 | 17.02 |
| **PPT** | **ROGA** | **97.62** | **26.19** |
| | AutoAgent | 73.81 | 19.05 |
| | OctoTools | 73.81 | 23.81 |
| | OWL | 80.95 | 21.43 |

expectation that the chart should be placed on the current sheet. SSC cannot easily detect such mismatches with latent, high-level user intent.

• **Subtle Variable Confusion.** This can occur during long-term tasks where the Persistent Symbolic Memory (PSM) becomes populated with numerous variables and objects with similar names or descriptions (e.g., `df_filtered`, `df_final`, `df_summary`). The agent might occasionally select the wrong variable for an operation. This error is particularly insidious because the generated tool can be both syntactically correct and semantically reasonable for the incorrectly chosen variable, making it difficult for SSC's automated validation to flag the logical mistake.

We believe these limitations are not flaws in the SSC mechanism itself, but rather inherent challenges of natural language ambiguity and complex state management in open-ended tasks. They represent an important and exciting direction for future research, potentially involving interactive clarification with the user or more advanced state representation techniques.

## C.3    PERFORMANCE BREAKDOWN BY FILE TYPE

To provide a more granular understanding of agent performance, we analyzed the results on the OSWorld benchmark by breaking them down across the three office file types: Word, Excel, and PowerPoint. As shown in Table 6, this detailed comparison not only confirms ROGA's superiority but also reveals how its paradigm effectively handles varying types of complexity where baseline agents fail. This detailed comparison yields several key insights.

• **Consistent Superiority Across All File Types of ROGA.** ROGA consistently and significantly outperforms all baseline agents across every file type, in terms of both execution and task success rate. This demonstrates the general robustness of our proposed paradigm.

• **Performance of ROGA Scales Predictably with Complexity.** Baseline agents exhibit erratic and unpredictable performance. For instance, OctoTools and OWL achieve their lowest success rates on Word tasks, which are structurally the simplest, yet perform better on more complex PPTX files. In stark contrast, ROGA 's performance scales predictably with the inherent complexity of the file types. Its success rate is highest on structurally simpler Word files (42.86%), followed by state-intensive Excel files (31.91%), and then the most complex multi-modal PPTX files (26.19%). This graceful degradation demonstrates that ROGA's paradigm provides a robust and principled way to manage increasing task complexity, whereas baselines fail unpredictably.

• **ROGA Demonstrates Exceptional Execution Reliability.** A critical finding is ROGA's near-perfect execution rate (100% on Word, 91.5% on Excel, 97.6% on PPTX). In contrast, baselines frequently fail to even generate executable code, especially on complex Excel tasks (e.g., Octo-Tools' 46.8% execution rate). This shows that ROGA's framework, particularly its **Situated Self-**

**Correction**, reliably produces syntactically correct and contextually valid tools. While baselines often fail at a fundamental level, ROGA successfully overcomes the stateful execution challenges.

# D    IMPLEMENTATION DETAILS

## D.1    PLANNER PROMPTS

The planner is the central cognitive component of ROGA, responsible for orchestrating the agent's decision-making process. Its behavior is guided by a set of carefully engineered prompts that enforce our proposed agent paradigm.

The system prompt of the planner establishes the agent's core identity and operational principles. It defines the agent's role, instructs it to strictly follow the Active World Modeling (AWM) paradigm by separating a dedicated comprehension phase, where the planner should iteratively build a world model before acting. The prompt details the available action space ('toolgen', 'toolexec', 'code-exec', 'done'), and specifies the required JSON output format.

The user prompt template provides the planner with the dynamic context required for each decision step. The context includes the original user instruction, the current state of the agent's world model (retrieved from the Persistent Symbolic Memory), the set of available capabilities (managed by the Dynamic Capability Evolution), and the recent code context. Together, these prompts guide the planner to iteratively probe the environment, build its understanding, and execute tasks in a structured, state-aware manner.

## D.2    TOOL VALIDATION WORKFLOW

Our Situated Self-Correction (SSC) mechanism is implemented as a dual-reflection tool generation process, including execution-based reflection for situated functional testing and semantic-based reflection for situated semantic consistency.

The tool validator is a key part of this mechanism. For each generated tool, the tool validator first executes it in a replica of the current sandbox (i.e., the shadow sandbox) for execution-based testing. If the execution process does not produce errors, the tool validator then uses LLMs to check its semantic correctness. The validator prompt provides an LLM with comprehensive context to perform a situated code review.

This context includes targeted functional capability of the tool, the code of the tool candidate, and a summary of the current state from the Persistent Symbolic Memory (e.g., relevant parts of the world model and available variables). The prompt then explicitly asks the LLM to assess whether the tool correctly implements the desired logic, uses the correct variables, and will functionally contribute to solving the sub-task as intended. The validator's feedback (a pass/fail decision along with suggested corrections) directly drives the Dynamic Capability Evolution process, determining whether a tool is promoted to the 'Active' state or sent back for further refinement.

## D.3    HYPERPARAMETER CONFIGURATION

This section details the key hyperparameters used for ROGA throughout our experiments. These settings were kept consistent across all benchmarks to ensure a fair comparison.

We set the temperature to 0.0 for all LLM calls. We set the maximum number of tokens for an LLM response to 32768. This provides sufficient length for the agent to generate complex code for tools and detailed reasoning steps without being prematurely cut off. The maximum reasoning steps of an agent are limited to 10 steps per task.

In ROGA, in a tool generation-validation loop, the agent attempts to refine the tool up to 3 times. If it still fails, the agent will reconsider its plan or try generating a new tool. A tool is moved to the 'Deprecated' state after 3 cumulative adaptation failures. During the planning phase, the agent retrieves the top-10 most relevant tools from its active set to consider for reuse.

Note that implementation details of ROGA are publicly available[2].

---

[2]https://github.com/morgen52/roga

