# OpenReview forum: "ROGA: Scaling Generalist Agents for Office Productivity Tasks via Tool Generation"
_ICLR.cc/2026/Conference — ICLR 2026 Poster_

### Official Review · Reviewer_akoz · 2025-10-27

**Soundness:** 3
**Presentation:** 3
**Contribution:** 2
**Rating:** 4
**Confidence:** 3

**Summary:**

The paper “ROGA: Scaling Generalist Agents for Office Productivity Tasks via Tool Generation” introduces a new framework, ROGA, to advance the paradigm of Automatic Tool Generation (ATG) for generalist agents operating in real-world office environments such as Excel, Word, and PowerPoint. The authors identify three major limitations of existing ATG agents: (1) poor handling of long file contexts that contain essential task details, (2) lack of shared state across tool executions, which prevents iterative modifications of the same object, and (3) inefficient tool reuse that leads to repeated generation errors. To address these, ROGA proposes four key innovations — (i) a Comprehension–Operation (Comp-Op) paradigm that separates task understanding from execution to capture fine-grained file details, (ii) a Dual-Reflection mechanism that performs both functional and semantic validation of generated tools, (iii) a State-Sharing Sandbox for consistent intermediate context management, and (iv) a Finite-State Machine (FSM) for tool lifecycle management that automates validation, reuse, and deprecation of tools. Across benchmarks including OSWorld, WindowsAgentArena (WAA), GAIA-Office, TableBench, and SheetCopilotBench, ROGA achieves significant improvements

**Strengths:**

This paper presents a well-motivated and empirically grounded contribution to the ATG research domain. The authors clearly identify critical pain points in current generalist agent paradigms and systematically address them through a thoughtfully designed architecture. The Comprehension–Operation paradigm formalizes reasoning separation, allowing agents to process long office files in a modular, interpretable manner. The Dual-Reflection mechanism provides a concrete and reproducible approach for improving tool reliability through iterative validation, while the State-Sharing Sandbox effectively enables multi-step tool coordination—an essential requirement for real-world automation tasks. The paper demonstrates methodological rigor with detailed benchmark coverage, using multiple baselines (AutoAgent, OctoTools, OWL, and SheetAgent) and both office and non-office domains. The inclusion of a well-structured ablation study offers clear evidence of how each module contributes to performance, revealing meaningful insights into which design choices matter most. Empirically, ROGA delivers strong and consistent gains across all benchmarks, notably outperforming specialized agents in spreadsheet reasoning tasks.

**Weaknesses:**

The paper lacks a deeper theoretical or analytical justification for why the proposed mechanisms—particularly dual-reflection and comprehension–operation separation—lead to better generalization and reduced reasoning errors. Although benchmarks and metrics are clearly listed, the paper does not provide implementation details for the planner’s decision policy, the functional testing procedure, or exact hyperparameter configurations for reflection or state management. The work mentions “shared memory Mt” and “transition function δ” but omits practical instantiation details, making it difficult for results to be replicated. Third, while ROGA achieves strong empirical results, the evaluation is mostly limited to office-centric benchmarks and one general reasoning dataset. It would be valuable to include cross-domain or zero-shot adaptation studies to demonstrate broader generalization and scalability claims. The cost analysis shows higher token consumption and reasoning steps, indicating that ROGA’s efficiency and scalability over very long tasks are not fully optimized; no discussion is provided on potential mitigation strategies.

**Questions:**

please address above questions in weakness

---

> ### Author Response · Authors · 2025-11-21
>
> Thank you for your thorough and insightful review. We are delighted that you recognized our work as a "well-motivated and empirically grounded contribution" and appreciated the methodological rigor of our evaluation and the systematic design of the ROGA architecture. Your constructive feedback regarding the need for deeper analytical justification, clearer implementation details for reproducibility, broader cross-domain evaluation, and a discussion of cost mitigation has been invaluable. We have revised our manuscript to address these points directly, and we will now respond to each of your concerns in detail.

---

> ### Author Response · Authors · 2025-11-21
> **Rebuttal to Weakness 1**
>
> > **Rebuttal to Weakness 1:** The paper lacks a deeper theoretical or analytical justification for why the proposed mechanisms—particularly dual-reflection and comprehension–operation separation—lead to better generalization and reduced reasoning errors.
>
> Thank you for your insightful feedback. We appreciate the opportunity to clarify the underground rationale behind our deign. We would like to highlight that the effectiveness of ROGA stems not from the engineered component in isolation, but from the fundamental paradigm shift and principle they represent. Specifically, we detail the analytical justification below.
>
> **1. Active World Modeling (AWM, implemented as comprehension–operation separation) Reduces Reasoning Errors by Grounding Actions in a Verified World Model.**
>
> The primary source of reasoning errors in prevailing agent paradigms when facing complex environments (like large office files) is **acting on incomplete or hallucinated information**. These paradigms passively attempt to perceive the entire environment, which is impossible due to context window limitations, forcing the agent to guess. AWM provides a principled solution to this problem of **partial observability**.
>
> **How it Reduces Errors:** AWM replaces passive perception with an active, iterative, meta-cognitive loop: **(1) Identify knowledge gap -> (2) Formulate specific question -> (3) Generate a comprehension tool to find the answer -> (4) Integrate the verified answer into the world model.** Instead of reasoning over a massive, unverified context, the agent reasons in small, verifiable steps. For example, it no longer assumes what is in a spreadsheet; it *knows* because it just ran a specific tool to check. This piece-by-piece construction of a world model from grounded facts directly eliminates reasoning errors that arise from making assumptions about a partially observed state.
>
> **2. Situated Self-Correction (SSC, implemented as Dual-Reflection) Reduces Errors and Enhances Generalization by Contextualizing Validation and Driving Evolution.**
>
> Prevailing paradigms often generate and validate tools in a decontextualized vacuum, leading to tools that are syntactically correct but fail in the specific context of a multi-step task. SSC addresses this by grounding the entire validation process in the agent's current state.
>
> **How it Reduces Errors:** SSC (implemented as dual-reflection) acts as a state-aware guardrail. The key innovation is its *situated* nature. Validation is not a generic check; it answers the question: "Does this tool work correctly and achieve the intended semantic goal *given the current state of my Persistent Symbolic Memory*?" By performing functional and semantic checks within a replica of the current state, SSC catches state-dependent errors (e.g., operating on an object that was modified three steps ago) *before* they corrupt the workflow. This directly prevents the cascading failures typical in long, stateful tasks.
>
> **How it Enhances Generalization:** SSC is the core engine of our **Dynamic Capability Evolution** paradigm. When a tool fails in a new situation, SSC doesn't just report failure; it provides precise, contextual feedback on *why* it failed in *that specific state*. This rich error signal drives the meta-learning process, allowing the agent to refine its capabilities. Over time, this iterative refinement makes the agent's skillset more robust and adaptable to a wider variety of situations. The agent generalizes its *skills*, not just its knowledge, by learning from experience across diverse contexts.
>
> In summary, the justification is not in the components themselves, but in the principles they embody. **Active World Modeling** ensures the agent reasons based on a grounded reality, not assumption, while **Situated Self-Correction** ensures its actions are valid within that evolving reality and that it learns from its mistakes. Together, ROGA create a more robust and adaptive agent that can generalize its problem-solving strategies to new, complex, and stateful environments, evidenced in office tasks.

---

> ### Author Response · Authors · 2025-11-21
> **Rebuttal to Weakness 2**
>
> > **Rebuttal to Weakness 2:** The paper does not provide implementation details for the planner’s decision policy, the functional testing procedure, or exact hyperparameter configurations, making it difficult for results to be replicated.
>
> Thank you for your valuable feedback. In our revised paper, we have added a new appendix section (Appendix "Implementation Details") dedicated to implementation details. This appendix provides the prompts, workflow, and other detailed implementations of our core mechanisms.
>
> Specifically, we provide the following clarifications, with details now available in the appendix:
>
> 1. **Implementations of Planner's Decision Policy:** In ROGA, the planner's policy is implemented via a set of carefully engineered system and user prompts that guide the LLM's reasoning. As detailed in the new appendix subsection "Planner Prompts", these prompts orchestrate the agent's behavior by providing it with:
>     - A core identity and instruction to follow the **Active World Modeling (AWM)** paradigm.
>     - The dynamic context for each step, including the user's goal, the current world model from the Persistent Symbolic Memory, and the set of available capabilities managed by Dynamic Capability Evolution.
>     - A clearly defined action space and the required JSON output format.
>
>     The detailed prompts are now provided in the appendix, making the decision-making logic explicit and replicable.
>
> 2. **Functional Testing Procedure:** Our **Situated Self-Correction (SSC)** mechanism is implemented as a dual-reflection generation process. We have clarified this workflow in the appendix under "Tool Validation Workflow." The process is as follows:
>     - **Step 1: Execution-Based Functional Testing.** When a new tool is generated, it is first executed within a "shadow sandbox", a replica of the current state of the Persistent Symbolic Memory. This step directly tests the tool's functional correctness in the current, specific context. If the tool executes without error, it proceeds to the next step.
>     - **Step 2: Prompt-Based Semantic Validation.** Only after a tool passes the functional test do we use a prompt-driven LLM for semantic validation. As shown in the appendix, we provide the LLM with the tool's code, description, and the current task context from the Persistent Symbolic Memory, to review its logic for semantic alignment.
>     This two-step process ensures that tools are both functionally executable and semantically correct in a situated manner.
> 3. **Hyperparameter Configurations:** We have added a dedicated subsection in the appendix, "Hyperparameter Configuration," that lists the crucial hyperparameters used in our experiments. This includes LLM-specific settings (e.g., temperature set to 0.0), framework-specific parameters such as the maximum reasoning steps per task (10), the maximum refinement attempts for a tool (3), the cumulative failure threshold for deprecating a tool (3), and the number of tools retrieved for consideration (top-k= 10). This detailed list provides the necessary parameters to replicate our experimental setup.
>
> We believe these additions to the appendix now offer a clear description of our implementation. Thank you again for helping us improve the completeness of our paper.

---

> ### Author Response · Authors · 2025-11-21
> **Rebuttal to Weakness 3**
>
> > **Rebuttal to Weakness 3:** The evaluation is mostly limited to office-centric benchmarks. It would be valuable to include cross-domain studies to demonstrate broader generalization.
>
> Thank you for this suggestion. We would like to respectfully clarify that our evaluation **already includes a cross-domain study** to validate ROGA's general-purpose nature, as detailed in **Section 4.4 ("Generalization on Non-Office Tasks")** and **Table 5** of our origin paper. This results demonstrates that while ROGA introduce a new agent paradigm capable of solving long-term, stateful tasks in partially observable environments, it would not lose its effectiveness on established, more general tasks like Math500.
>
> Following your suggestion, we have added additional cross-domain studies to demonstrate its effectiveness. We utilize **MMLU-Pro**, a challenging multi-task understanding benchmark, to test a different and critical capability of generalization: knowledge-intensive reasoning. As shown in our revised paper (Table 5, also shown below), demonstrates strong and competitive performance across both domains. The results further indicate that ROGA's design successfully enables it to tackle complex, stateful tasks while maintaining performance on a broad spectrum of general-purpose reasoning benchmarks.
>
> | Method | Math500 (Exec@1) | Math500 (Pass@1) | MMLU-Pro (Exec@1) | MMLU-Pro (Pass@1) |
> | --- | --- | --- | --- | --- |
> | AutoAgent | 84.00 | 54.00 | 100.00 | 78.00 |
> | OctoTools | 100.00 | 70.00 | 97.00 | 64.00 |
> | OWL | 99.00 | 71.00 | 93.00 | 65.00 |
> | ROGA | 99.00 | 70.00 | 100.00 | 76.00 |

---

> ### Author Response · Authors · 2025-11-21
> **Rebuttal to Weakness 4**
>
> > **Rebuttal to Weakness 4:** The cost analysis shows higher token consumption and reasoning steps, but no discussion is provided on potential mitigation strategies.
>
> Thank you for this valuable feedback. Here we would like to clarify that the higher initial cost is a **design trade-off for robustness**, but ROGA features a **built-in mitigation mechanism** and could inspire some potential path for **future cost optimization**. We have revised Section 4.2 to provide a more in-depth cost analysis.
>
> 1. **Built-in Mitigation: Amortized Cost via Dynamic Capability Evolution (DCE)**
>
>     Our primary cost mitigation strategy is already integral to the framework: **Dynamic Capability Evolution (DCE)**. This mechanism operates on an "invest once, use many times" principle.
>
>     - **The Cost:** The high cost is incurred upfront when a *new* capability is generated and rigorously validated.
>     - **The Mitigation:** Once validated, this capability becomes a cheap, reusable asset. The cost of retrieving and executing a validated tool is a fraction of the original generation cost. This amortizes the initial investment over time, making the agent more efficient as it encounters similar tasks.
>     - **Evidence:** Our ablation study (Section 4.2) supports this. Removing the capability lifecycle management led to a **1.54x increase in reasoning steps**, demonstrating that our reuse mechanism is already effective at reducing long-term operational costs.
> 2. **Future Mitigation Strategies**
>
>     Building on your suggestion, we have identified several promising directions for future work to further optimize cost:
>
>     - **Cost-Aware Planning:** We can enhance the Planner to perform a cost-benefit analysis. Before generating a new tool, it could estimate the cost and compare it against the potential benefit. This would allow the agent to make more strategic, resource-aware decisions.
>     - **Adaptive World Modeling:** The Active World Modeling process can be made more efficient. The agent could learn to prioritize which parts of the environment are most critical to probe first, based on past experience with similar tasks. This would reduce the number of iterative comprehension cycles needed, directly lowering token consumption.
>
> In summary, ROGA's current cost profile reflects its focus on achieving correctness where other methods fundamentally fail. Our framework’s design inherently supports long-term efficiency through capability reuse, and we have discussed the potential roadmap for further cost optimization. Thank you for helping us to clarify this important aspect.

---

> ### Author Response · Authors · 2025-11-27
> **Request for Reviewer's Feedback**
>
> Dear Reviewer akoz,
>
> Thank you again for your insightful review.
>
> We want to follow up on our response and ask if there is any further information we can provide to address your concerns. Our goal is to ensure the paper is as robust and clear as possible, and we welcome any additional thoughts you might have as the discussion window continues.
>
> We truly appreciate your engagement and hope to merit your positive recommendation.
>
> Sincerely,
>
> Authors

---

### Official Review · Reviewer_SvUL · 2025-10-30

**Soundness:** 3
**Presentation:** 2
**Contribution:** 3
**Rating:** 4
**Confidence:** 4

**Summary:**

This paper introduces ROGA, a generalist agent framework designed to overcome limitations of current Automatic Tool Generation (ATG) agents in open-ended office productivity tasks. The authors identify three key shortcomings of existing ATG agents: inadequate handling of long file contexts, lack of context sharing across tools, and inefficient tool reuse. ROGA addresses these via a comprehension-operation reasoning paradigm, dual-reflection tool generation, a state-sharing execution sandbox, and finite-state machine-based tool lifecycle management. Evaluations on office benchmarks show that ROGA significantly outperforms existing ATG agents, while preserving performance in non-office domains.

**Strengths:**

1. The paper includes a well-designed motivation study that clearly illustrates the limitations of existing generalist agent frameworks, particularly in office environments.

2. Performance: ROGA demonstrates strong and consistent performance on office productivity tasks, and the authors show that it does not deteriorate on non-office domains, supporting its generalizability.

3. The overall writing is generally clear, and the framework components are explained in a structured and understandable manner.

**Weaknesses:**

1. In the motivation study, the authors compare ATG agents to the domain-specific SheetAgent to highlight performance gaps. However, in the main experiments, ROGA is not compared to domain-specific agents (except in the spreadsheet-only analysis). Including such comparisons for other office tasks (e.g., Word or PowerPoint) would better contextualize ROGA’s advancements.

2. The paper does not clearly motivate why office productivity tasks are particularly important or representative as a testbed for generalist agents. A stronger justification in the intro—for instance, their complexity, real-world impact, or ubiquity—would enhance the significance of the chosen domain.

3. While ablations show that each component contributes to performance, it remains unclear whether ROGA’s gains stem primarily from its general agentic design or from better, office-specific tooling and context handling. The framework is presented as general, yet its advantages are most pronounced in office tasks.

**Minor Issues**

Line 478: "environments To address them" – missing a period.

Citation formatting is inconsistent in several places, e.g., in line 113 "(Cai et al., 2024; Wolflein et al., 2025; Qian et al., 2023) aims to reduce..." should use consistent punctuation and style.

**Questions:**

1. The framework components (e.g., Comprehension–Operation, State-Sharing Sandbox) appear general and domain-agnostic. Why, then, does ROGA significantly improve performance on office tasks but show only comparable (not superior) results on non-office tasks like math? What aspects of the design are specifically tailored to office environments?

2. Could the authors provide more insight into whether the performance gains are due to better context management and tool generation for office files, or due to the agentic framework itself?

---

> ### Author Response · Authors · 2025-11-21
>
> Thank you for your insightful review and for recognizing the strengths of our work, including the well-designed motivation study and ROGA's strong performance. We appreciate your constructive feedback, which has helped us to further clarify the significance of our chosen domain and the generality of our framework's contributions. We will now address your specific comments and questions in detail.

---

> ### Author Response · Authors · 2025-11-21
> **Rebuttal to Weakness 1**
>
> > **Rebuttal to Weakness 1**: The comparison with domain-specific agents for Word and PowerPoint.
>
> Thank you for this insightful comment. We would like to respectfully clarify that our experimental setup already includes such a comparison, as one of our key baselines, OWL, functions as a system composed of such predefined, specialized agents for tasks.
>
> Specifically, OWL (Hu et al., 2025) is a multi-agent system that assigns specific tasks to predefined, specialized agents. For office automation, OWL employs dedicated agent types, including an `ExcelAgent` for spreadsheet tasks and a `DocumentAgent` for processing Word and PowerPoint files. Therefore, our comparison against OWL on the OSWorld, WAA, and GAIA-Office benchmarks, which include Word and PowerPoint tasks, is effectively a comparison against a strong, domain-specific approach.
>
> As the results from our main experiments show, ROGA consistently and significantly outperforms the specialized agent system of OWL across these benchmarks. For instance, on the OSWorld benchmark, ROGA improves the Pass@1 rate by 15.46 percentage points over OWL, and on the WAA benchmark, the improvement is 9.52 percentage points. This outperformance of a specialized system highlights the significant advancements of ROGA's paradigm.
>
> | Benchmark | Agent | Exec@1 (%) | Pass@1 (%) |
> | --- | --- | --- | --- |
> | **OSWorld** | OWL | 79.09 | 16.36 |
> |  | **ROGA** | **95.45** | **31.82** |
> | **WAA** | OWL | 76.19 | 19.05 |
> |  | **ROGA** | **97.62** | **28.57** |
> | **GAIA-Office** | OWL | 92.31 | 73.08 |
> |  | **ROGA** | **96.15** | **76.92** |
>
> To make this point more explicit for readers, we add a clarification in the experimental setup section of our revised manuscript, highlighting that OWL serves as a strong domain-specific baseline for office tasks. Thank you again for helping us improve the clarity of our paper.

---

> ### Author Response · Authors · 2025-11-21
> **Rebuttal to Weakness 2**
>
> > **Rebuttal to Weakness 2:  The motivation for choosing office productivity tasks as a testbed for generalist agents.**
>
> Thank you for this valuable suggestion. We do provide some initial motivation in the introduction (Section 1), where we mention that these tasks have "broad practical impact", "support workflows that consume billions of user hours each day", and require "extended reasoning steps." However, we acknowledge this can be strengthened.
>
> In our revision, we follow your suggestion to more explicitly and forcefully clarify that these tasks are the **ideal crucible** for testing generalist agents and are uniquely challenging because they combine:
>
> 1. **Massive real-world impact and ubiquity**, making them a critical target for automation. The office workflows "consume billions of user hours each day" and form the backbone of countless business and academic activities. The success in office automation has a direct and massive impact on productivity, making it a critical target for generalist agents aiming for real-world utility.
> 2. **Inherent complexity** (long-term, stateful interaction, and partially observable file contexts), which directly exposes the fundamental flaws of prevailing ATG agent paradigms that assume fully observable environment, passively perceive the environment, and involve only simple, stateless, short-term action.
>
> We also highlight that any agent claiming to be "generalist" must be proficient in this domain. If an agent can only handle simple API calls or web navigation but fails at common office tasks, its generalist capabilities are fundamentally incomplete. By making these points more prominent in the introduction, we hope to better frame the significance of our work and the chosen domain.

---

> ### Author Response · Authors · 2025-11-21
> **Rebuttal to Weakness 3**
>
> > **Rebuttal to Weakness 3:  While ablations show that each component contributes to performance, it remains unclear whether ROGA’s gains stem primarily from its general agentic design or from better, office-specific tooling and context handling. The framework is presented as general, yet its advantages are most pronounced in office tasks.**
>
> Thank you for your insightful question. We agree that ROGA’s advantages are most pronounced in office tasks. This is by design, as our **primary research goal was to advance the generalist agent paradigm to handle long-term, stateful tasks**, for which office automation is a key and challenging example. The core innovations of ROGA are indeed general agentic principles, and office tasks simply serve as the ideal environment to demonstrate their value over previous paradigms.  We have revised our paper in Section 1, 3, 4.2 to make this point clearer.
>
> Here is a point-by-point clarification:
>
> 1. **Our Core Contributions Are General Agentic Principles.** We first  identify three core limitations in current agent paradigms, including failure to build a world model, memory-less execution, and static capability generation. These are undamental flaws of the prior agent paradigm, not just in office tasks. Our solutions are correspondingly general principles:
> - **Active World Modeling** is a general solution for any task where the environment is too large or complex to fit in a context window (e.g., navigating large codebases, complex websites, or entire file systems). Office files are just one example of such a partially observable environment.
> - **Persistent Symbolic Memory** is a general mechanism for any long-term task that requires tracking state across multiple, dependent actions. This is critical for iterative data analysis, multi-step software configuration, and many other complex workflows, not just editing a document.
> - **Dynamic Capability Evolution** is a general meta-learning framework that allows an agent to adapt and improve its skills over time. This principle is not tied to office tools and is applicable to any agent that operates in an evolving environment where similar sub-tasks recur.
> 2. **Office Tasks Serve as a Challenging Testbed, Not a Limitation.**
>
>     We chose office tasks precisely because they are a domain where the weaknesses of existing generalist agents are most apparent. Simpler, stateless tasks (like many math problems) do not sufficiently stress an agent's ability to handle long contexts, track state, or capability evolution capabilities. Therefore, the performance gap between ROGA and other agents is naturally smaller on such tasks.
>     The significant performance gains on office tasks do not indicate that ROGA is an "office-specific" agent. Rather, they demonstrate that ROGA's general principles are highly effective at solving the very class of problems (long-term, stateful interaction) that have been a major bottleneck for the practical application of generalist agents.
>
> 3. **Empirical Evidence Confirms Generalization.**
>
>     To explicitly validate that ROGA's design maintains its general-purpose nature, we evaluated it on the non-office Math500 and MMLU-Pro (in revised paper) benchmark. As shown in Table 5, ROGA achieves performance on par with the best-performing baseline agents. This result is crucial, as it demonstrates that our framework successfully extends the capabilities of generalist agents into a new, complex domain **without sacrificing** their performance on established, more general tasks.
>
> In summary, ROGA's framework is fundamentally general. The pronounced gains in office tasks highlight that this domain is a powerful stress test for agentic capabilities, and our proposed paradigm effectively addresses these challenges. We believe this work demonstrates a viable path to scale generalist agents to more realistic and complex scenarios.

---

> ### Author Response · Authors · 2025-11-21
> **Response to Minor Issues**
>
> Thank you for carefully catching these details. We have addressed all the minor issues you proposed.

---

> ### Author Response · Authors · 2025-11-21
> **Response to Question 1**
>
> > **Response to Question 1: The framework components appear general and domain-agnostic. Why, then, does ROGA significantly improve performance on office tasks but show only comparable results on non-office tasks like math? What aspects of the design are specifically tailored to office environments?**
>
> Thank you for this insightful question. We clarify that the performance difference (between office and mathematic tasks) arises NOT because ROGA is tailored *to* office applications, but because it is designed to solve a **class of problems** that are long-term, stateful, and with partially observable environments, which are characteristic of office environments but largely absent in simpler, stateless tasks like the math problems we tested. We have revised our paper in Section 4.2 to make this point clearer.
>
> Here is a more detailed breakdown:
>
> 1. **The Mismatch Between Problem Characteristics and Solution Features**
>
>     The key is that different benchmarks test for different agent capabilities.
>
> - **Office Tasks** (e.g., WAA, OSWorld) are defined by:
>     - **Partial observability:** The agent cannot see the entire file (e.g., a 50-page Word document or a 10-sheet Excel file) in a single context window. It must actively explore to build an understanding.
>     - **Statefulness:** Actions are iterative and dependent. An action in step 5 (e.g., creating a chart) relies on the state created by an action in step 2 (e.g., formatting a data table). The state of the file object must be maintained across steps.
>     - **Long-term, recurring sub-tasks:** Similar operations (e.g., finding specific text, applying a format) occur repeatedly, making tool reuse highly beneficial.
> - **Mathematics Tasks**, in contrast, are largely:
>     - **Fully observable:** The entire problem is a short text string that fits easily into the context window.
>     - **Stateless:** The solution typically requires a single, independent tool call (e.g., executing a Python script with a `math` library). The state does not need to be tracked across multiple, dependent tool executions.
>     - **Non-recurring tool generation:** A single, well-defined calculator or code interpreter tool is often sufficient. There is little need for dynamic tool evolution within a single problem.
> 2. **Why ROGA's Components Are "Activated" by Office Tasks**
>
>     ROGA's core innovations directly address the challenges of office tasks, but these challenges are not present in the math benchmark.
>
> - **Active World Modeling (implemented as Comprehension-Operation):** This is our solution for **partial observability**. It allows the agent to iteratively probe a large file to build a mental model. For a short math problem that is fully visible, this capability is not needed and remains largely dormant.
> - **Persistent Symbolic Memory (implemented as State-Sharing Sandbox):** This is our solution for **statefulness**. It ensures that when the agent modifies a file object in one step, that modified state is available for the next step. For a stateless math problem, there is no evolving object state to track, so this component provides no advantage.
> - **Dynamic Capability Evolution (implemented as Tool Lifecycle):** This is our solution for efficient handling of **recurring sub-tasks**. It allows the agent to reuse and refine tools. In office tasks, this avoids costly and error-prone regeneration. In math tasks, where a single tool use is often enough, the benefit is minimal.
> 3. **The Math Result is a Confirmation of Generality, Not a Limitation**
>
>     The fact that ROGA performs on par with other agents on Math500 is a crucial result that validates our approach. Our goal was not to build a superior math solver, but to create a more powerful generalist agent that could tackle a new, harder class of problems **without sacrificing on existing ones**.
>
>
> In summary, **no aspect of ROGA's design is hard-coded for "office."** Instead, ROGA is designed for **"long-term, stateful interaction in partially observable environments."** Office tasks are the typical example of this problem class, which is why the performance gains are so dramatic there. The math tasks, being stateless and fully observable, simply do not require the advanced features ROGA offers, leading to comparable performance.

---

> ### Author Response · Authors · 2025-11-21
> **Response to Question 2**
>
> > **Response to Question 2: Could the authors provide more insight into whether the performance gains are due to better context management and tool generation for office files, or due to the agentic framework itself?**
>
> Thank you for this excellent follow-up question. We clarify that the performance gains are due to the **agentic framework** itself, which in turn enables superior context management and tool generation capability in long-term, stateful environments like office tasks. We do not build better office-specific tools; we build a better framework that is capable of better generating and managing them.
>
> Here we show some in-depth insights.
>
> 1. **"Better Context Management" is a direct result of "Active World Modeling."**
> Prevailing agents fail at context management in office tasks because they passively try to ingest an entire large file as a whole, which is impossible. Our solution is not a clever parsing trick. Instead, our **Active World Modeling** paradigm changes the agent's fundamental behavior from passive perception to active, iterative probing. The framework compels the agent to ask questions about the environment (e.g., "What sheets are in this workbook?") and then generate *comprehension tools* to find the answers. This is a framework-level innovation in how the agent interacts with any partially observable world, not a specific technique for office files.
> 2. **"Better Tool Generation" is a direct result of "Dynamic Capability Evolution" and "Persistent Symbolic Memory."**
> - Prevailing agents generate tools in a stateless vacuum, which often fails in office tasks where actions are sequential and interdependent. Our framework solves this. The **Persistent Symbolic Memory (PSM)** ensures that every tool is generated and validated within the *current, evolving state* of the task.
> - For example, imagine this scenario:  when generating a tool to create a chart, ROGA knows from PSM that a specific data table has already been created and formatted in a previous step. The tool is generated with this state awareness. This state-aware generation, combined with our **Dynamic Capability Evolution** which reuses and refines proven tools, is what leads to "better tools." It is not the final tool code that is novel, but the state-aware, evolutionary *process* of creating it, which is an integral part of the agentic framework.
>
> In essence, the "better tooling and context management" we observe are **symptoms and results** of a more advanced underlying agentic framework. The framework's principles are general, but their benefits (including better context handling and stateful tool generation) are most visible and impactful when stress-tested in complex domains like office automation. This is precisely why the framework shows significant gains there, while its advanced features are not required for simpler, stateless tasks like the math problems.
>
> To quantitatively demonstrate how each component of our agentic framework contributes to these capabilities, we present a summary of our ablation study from the paper (Table 2). The results on the OSWorld benchmark clearly show that removing any of these general-purpose framework components leads to a significant performance drop, even in a diverse environment beyond just office files.
>
> | Configuration | OSWorld (Pass@1) | Performance Drop | Analysis: Why it Fails |
> | --- | --- | --- | --- |
> | **ROGA (Full Framework)** | **31.82%** | - | Baseline performance with all agentic components. |
> | `- w/o Active World Modeling` | 20.00% | **-11.82** | **Fails to build a coherent world model.** The agent acts on incomplete understanding. |
> | `- w/o Persistent Symbolic Memory` | 19.09% | **-12.73** | **Loses state continuity.** Actions become isolated and fail in multi-step sequences. |
> | `- w/o Dynamic Capability Lifecycle` | 29.09% | **-2.73** | **Repeatedly regenerates tools, increasing error risk** and overhead (1.54x more steps). The performance drop is mitigated by the Situated Self-Correction mechanism, which enhances regeneration accuracy. |
> | `- w/o Situated Self-Correction` | 23.64% | **-8.18** | **Generates faulty tools.** Without state-aware validation, tools have more errors. |
>
> As the data shows:
>
> - Removing **Active World Modeling** causes a massive **11.82-point drop** in task success. This directly proves that without the framework's ability to actively probe the environment, the agent cannot form an accurate understanding, leading to failure.
> - Removing **Persistent Symbolic Memory** results in the largest drop of **12.73 points**. This highlights that maintaining state is the most critical factor for success in these long-horizon tasks. Without this framework component, the agent is fundamentally incapable of performing iterative operations.

---

> ### Author Response · Authors · 2025-11-27
> **Request for Reviewer's Feedback**
>
> Dear Reviewer SvUL,
>
> We sincerely appreciate your thoughtful feedback and the time you've dedicated to our manuscript.
>
> We are just writing to see if our rebuttal was clear and satisfactorily answered your questions. Your comments have been incredibly helpful for improving the paper. We remain on standby to address any other points or provide more detail as needed.
>
> We value your expert opinion and hope our efforts meet with your approval.
>
> Sincerely,
>
> Authors

---

### Official Review · Reviewer_jaLz · 2025-11-01

**Soundness:** 2
**Presentation:** 3
**Contribution:** 2
**Rating:** 4
**Confidence:** 3

**Summary:**

The paper presents ROGA, a framework designed to enhance automatic tool generation (ATG) agents for office productivity tasks such as Excel, Word, and PowerPoint automation. The authors argue that existing ATG agents struggle in realistic, long-context environments due to three limitations: poor handling of extended file contexts, lack of context sharing across tool invocations, and inefficient tool reuse.

To overcome these issues, ROGA introduces four components: a Comprehension–Operation paradigm to separate file understanding from action, a Dual-Reflection mechanism for functionally and semantically validated tool generation, a State-Sharing Sandbox for maintaining shared intermediate states, and a Finite-State Machine for managing tool lifecycles and reuse.

Extensive experiments across benchmarks such as OSWorld, WindowsAgentArena, GAIA-Office, TableBench, and SheetCopilotBench show that ROGA achieves substantial improvements over prior ATG agents, even matching or slightly exceeding domain-specific agents in spreadsheet tasks. The paper is well-structured, clearly written, and tackles a problem of growing real-world importance in automating office workflows.

This paper demonstrates strong engineering, clear motivation, and compelling empirical validation in an area of genuine practical importance. However, the core contributions lack sufficient methodological novelty to justify acceptance at a premier venue focused on conceptual advances. The main components, such as reading-then-acting, reflective testing, stateful execution, and lifecycle management, are well-executed but largely incremental adaptations of existing principles.

While ROGA’s performance gains are notable, they likely stem from implementing an architecture that correctly integrates established best practices rather than from introducing a fundamentally new paradigm for ATG agents. The high computational overhead, limited analysis of failure modes, and modest absolute success rates further temper enthusiasm.

Overall, this work is a solid systems contribution that convincingly identifies the architectural elements required for robust ATG performance in realistic office settings. With deeper algorithmic innovation or analytical rigor, it could evolve into a strong submission in future iterations.

**Strengths:**

Clear Motivation and Relevance:
The paper identifies an important and under-explored challenge, the adaptation of generalist ATG agents to realistic office environments that require multi-step reasoning and persistent file manipulation. The motivation study effectively quantifies the performance gap (up to 27.43%) between existing ATG agents and specialized systems, making a strong case for the work’s necessity.

Clarity and Presentation:
The writing is lucid and well-organized, with helpful figures and a formal task formulation. The overview diagram (Figure 1) and formal decision-process notation help convey the architecture’s structure.

Comprehensive Evaluation:
The experiments are broad in scope, spanning several public benchmarks, and the ablation study convincingly shows that each ROGA component contributes meaningfully to performance. Including Math500 to assess generalization beyond office tasks is a commendable touch.

Empirical Strength:
The reported results are impressive. ROGA delivers consistent improvements on all office benchmarks, and even outperforms the specialized SheetAgent in spreadsheet tasks. This demonstrates that the proposed system is robust, practically effective, and scalable to realistic workloads.

Practical Impact:
The focus on office task automation has major applied significance. Demonstrating tangible improvements in this space is valuable for both research and industry audiences.

**Weaknesses:**

While empirically strong, the paper’s technical novelty appears limited, and several of its key mechanisms resemble established engineering patterns:

The Comprehension–Operation paradigm closely parallels the long-standing plan-then-act or read-then-execute architectures common in agent design, making its conceptual contribution modest.

The Dual-Reflection mechanism essentially performs test-and-validate loops (shadow execution and code review), a well-known reliability technique in program synthesis and software testing.

The State-Sharing Sandbox is effectively a transactional execution buffer that maintains shared state, standard practice in interactive software systems.

The Finite-State Machine for tool lifecycle management resembles a conventional resource cache with validation and eviction logic.

In short, the framework’s advances seem to stem more from careful system integration than from new algorithmic ideas. Moreover, ROGA’s higher computational cost (3–4× token usage over baselines) raises concerns about scalability and efficiency, particularly since absolute success rates on some benchmarks (e.g., 31.82% Pass@1 on OSWorld) remain low.

Finally, the analysis would benefit from deeper insight into why the system works, for instance, what specific types of context or reflection yield measurable gains, and which failure modes persist. The absence of qualitative or statistical analyses limits interpretability and reproducibility.

**Questions:**

How exactly does the Comprehension–Operation separation improve results beyond simply allowing more reasoning steps? Could equivalent performance be achieved by giving baselines comparable computation budgets?

In the Dual-Reflection mechanism, how often does semantic validation detect issues missed by functional testing, and could examples illustrate these cases?

The paper shows that ROGA uses significantly more tokens than baselines. What fraction of this overhead comes from comprehension versus reflection, and how does this affect cost-benefit trade-offs?

What are the main failure cases (e.g., comprehension errors, tool misuse, sandbox inconsistencies), and do they suggest directions for further refinement?

Given that the improvements on TableBench and SheetCopilotBench are modest and the gains do not extend to Math500, how generalizable is ROGA beyond the office domain?

---

> ### Author Response · Authors · 2025-11-21
>
> Thank you for your detailed and constructive review. We are delighted that you recognized the empirical strength, clear motivation, and practical relevance of our work. We particularly value your insightful critique regarding the paper's conceptual novelty, as it has pushed us to significantly sharpen our core contributions. Your feedback has been invaluable in helping us revise the manuscript to more clearly articulate the fundamental paradigm shift that ROGA represents, distinguishing it from a simple integration of established engineering patterns. We will now address your concerns and questions in detail.

---

> ### Author Response · Authors · 2025-11-21
> **Rebuttal to Weakness 1: On Technical Novelty and Parallels to Established Patterns**
>
> Thank you for your insightful critique. We appreciate you pointing out that our original presentation failed to adequately articulate the core conceptual novelty of our work. Your feedback has been invaluable in helping us sharpen our message. We have undertaken a significant revision of the paper—particularly in the Abstract, Introduction, and Methodology sections—to move beyond a component-level description and instead articulate the fundamental *paradigm shift* that ROGA represents.
>
> We agree that at a surface level, our components have analogues in software engineering. However, our core contribution is not the invention of these mechanisms in isolation, but rather **the identification of fundamental flaws in prevailing agent paradigms and the proposal of a new, coherent agent paradigm that addresses them through a novel synthesis of algorithmic principles.**
>
> Our revised manuscript now frames ROGA's contributions as three foundational algorithmic innovations that constitute a *paradigm shift* for agents operating in long-term, stateful office environments:
>
> 1. **From prior "Plan-Then-Act" to Active World Modeling:** In contrast to the "plan-then-act" approach that passively understands a fully observable environment, the implemented "Comprehension phase" represents **Active World Modeling**. Prevailing paradigms assume a fully observable environment that can be "read" in one go. This fails in realistic settings with long, partially observable contexts (like complex office files). Active World Modeling is not a simple one-shot "read" step. It is a **meta-cognitive, iterative process where the agent actively probes the partially observable environment by dynamically generating specialized *comprehension tools* to build its own world model piece by piece.** This active, iterative construction of a world model from incomplete information is our core conceptual advance, going far beyond standard read-then-execute architectures.
> 2. **From prior "Stateless Tool Execution" to Persistent Symbolic Memory:** The sandbox purposed within our paradigm is more profound than that in the prior interactive software systems. It indeed represents a **Persistent Symbolic Memory**. Current agent paradigms suffer from a "memory-less execution model" where stateless tool calls loss temporal details. Our Persistent Symbolic Memory is not just a temporary buffer; it is an **explicit, structured symbolic state ledger that tracks the evolution of the world state across long action chains.** It provides symbolic handles to objects, enabling true temporal reasoning and stateful interaction (e.g., modifying the same dataframe object across ten sequential tool calls). This directly addresses the fundamental flaw of memory-less execution, which is a critical limitation for current agent paradigm to solve any complex, iterative task.
> 3. **From prior "Caching" to Dynamic Capability Evolution:** You astutely compared our FSM to a resource cache. However, it is actually an implementation of a novel agent evolution paradigm, **Dynamic Capability Evolution.** Prevailing paradigms are stuck in a "static capability generation" model, forcing redundant, one-shot tool generation. Our framework treats the agent's skillset as a dynamic entity that undergoes meta-learning for various stateful environments.
>     - The **Finite-State Machine** provides the **formalism for this evolutionary process**, modeling a capability's entire lifecycle from generation to refinement and retirement.
>     - The validation mechanism, now termed **Situated Self-Correction**, acts as the **core engine for this evolution**. It is not a generic test-and-validate loop. Its novelty lies in being *situated*—validation occurs within the context of the agent's current Persistent Symbolic Memory. It answers the question: "Does this capability work correctly *given the current state of my world model and action history*?" This context-aware feedback drives the evolution, allowing the agent to learn and adapt its capabilities over time.
>
> In summary, we have revised the paper to argue that the novelty of ROGA lies not in inventing transactional buffers or testing loops from scratch, but in identifying the deep-seated limitations of current agent paradigms and architecting a new one—instantiated in ROGA—that solves them. Our contribution is the conceptual framework of **Active World Modeling, Persistent Symbolic Memory, and Dynamic Capability Evolution**, which together enable agents to effectively tackle long-term, stateful tasks where previous paradigms have failed.
>
> We believe these revisions, directly inspired by your feedback, now more clearly articulate our paper's core scientific contributions and demonstrate that ROGA is more than the sum of its parts—it is an instance of a new, more powerful agent paradigm. We thank you again for pushing us to clarify our thinking and improve the paper.

---

> ### Author Response · Authors · 2025-11-21
> **Rebuttal to Weakness 2: Costs and Performance**
>
> Thank you for raising this critical point about cost and performance. We agree this is a crucial trade-off. We respectfully argue that ROGA’s higher cost is a necessary and efficient investment in a superior reasoning paradigm, and that the performance actually represents a significant breakthrough on previously intractable problems. In short, the increased cost directly funds our **Active World Modeling** paradigm, preventing the "high-cost failures" common in baselines that act on flawed assumptions. The performance result on OSWorld of ROGA is a **near-doubling of the success rate** over the prior methods. ROGA, our proposed generalist agent, has even **outperforms** a highly-specialized, SOTA agent (SheetAgent) in its own domain. These results highlight the extreme difficulty of the tasks and the effectiveness of our new approach.
>
> We provide a more detailed breakdown of these two points in our responses to your subsequent questions.
>
> - For a detailed analysis of **why the higher cost enables a more effective reasoning process rather than just brute-force computation**, please see our response to your Question 1.
> - For a deeper discussion on **why these performance gains are a significant breakthrough**, please see our response to your Question 5.

---

> ### Author Response · Authors · 2025-11-21
> **Rebuttal to Weakness 3**
>
> > **Rebuttal to Weakness 3: The analysis would benefit from deeper insight into why the system works, for instance, what specific types of context or reflection yield measurable gains, and which failure modes persist. The absence of qualitative or statistical analyses limits interpretability and reproducibility.**
>
> Thank you for your insightful feedback. We appreciate the opportunity to offer an in-depth analysis of why ROGA works. We respectfully clarify that ROGA’s gain stems from its foundational agentic paradigm, which transforms how the agent perceives, remembers, and acts in long-term, stateful environments. Below, we provide the specific qualitative insights and quantitative analysis following you requests.
>
> 1. **Qualitative Insight.**
>
> The core innovations of ROGA are designed to solve fundamental failure modes of previous agent paradigms in long-term, stateful, partially-observable tasks.
>
> - **Active World Modeling (AWM) Prevents Failures from Partial Observability.**
>
>     Current agents acting on incomplete or hallucinated context because they cannot perceive large environments (like an entire Excel file) as a whole due to context window limitations. Instead of passively reading, AWM actively probes the environment. It replaces risky assumptions with verified facts, ensuring actions are grounded in an accurate understanding of the world.
>
> - **Persistent Symbolic Memory (PSM) Prevents Failures from Memory-less Execution.**
>
>     Stateless tool calls in current agent paradigm can break the chain of causality in iterative tasks (e.g., modifying the same object over multiple steps). PSM provides a structured state ledger that tracks the evolution of objects across actions, enabling true stateful interaction. In this case, ROGA can track the *current* state of the world, preventing it from acting on outdated information and enabling complex, multi-step operations.
>
> - **Dynamic Capability Evolution (DCE) Prevents Failures from Static, Redundant Tool Generation.**
>
>     Agents repeatedly and inefficiently generating tools from decontextualized vacuum, leading to more errors and no long-term learning. DCE reframes tool generation as a meta-learning process where the agent reuses, refines, and adapts its own capabilities. Its evolution is driven by **Situated Self-Correction**, a process that validates tools against the *current state* in PSM. This catches context-dependent errors and drives adaptation, creating a more robust and efficient agent.
>
> 2. **Quantitative Analysis**
>
> To quantitatively demonstrate how each component of our agentic framework contributes to these capabilities, we present a summary of our ablation study from the paper (Table 2). The results on the OSWorld benchmark clearly show that removing any of these general-purpose framework components leads to a significant performance drop, even in a diverse environment beyond just office files.
>
> | Configuration | OSWorld (Pass@1) | Performance Drop | Analysis: Why it Fails |
> | --- | --- | --- | --- |
> | **ROGA (Full Framework)** | **31.82%** | - | Baseline performance with all agentic components. |
> | `- w/o Active World Modeling` | 20.00% | **-11.82** | **Fails to build a coherent world model.** The agent acts on incomplete understanding. |
> | `- w/o Persistent Symbolic Memory` | 19.09% | **-12.73** | **Loses state continuity.** Actions become isolated and fail in multi-step sequences. |
> | `- w/o Dynamic Capability Lifecycle` | 29.09% | **-2.73** | **Repeatedly regenerates tools, increasing error risk** and overhead (1.54x more steps). The performance drop is mitigated by the Situated Self-Correction mechanism, which enhances regeneration accuracy. |
> | `- w/o Situated Self-Correction` | 23.64% | **-8.18** | **Generates faulty tools.** Without state-aware validation, tools have more errors. |
>
> This analysis clearly shows that **Persistent Symbolic Memory** and **Active World Modeling** provide the largest measurable gains, as they address the most critical failures of memory-less execution and partial observability.
>
> We have integrated these qualitative and quantitative analyses into the revised manuscript in Section 1, 3, and 4.2 to enhance its interpretability. Thank you again for your constructive guidance!

---

> ### Author Response · Authors · 2025-11-21
> **Response to Question 1**
>
> > **Response to Question 1:** How exactly does the Comprehension–Operation separation improve results beyond simply allowing more reasoning steps? Could equivalent performance be achieved by giving baselines comparable computation budgets?
>
> Thank you for this excellent question. We respectfully clarify that the performance gain of ROGA is NOT a simple function of "more steps." It stems from a fundamental shift in the *paradigm* of the reasoning process. Giving baselines a comparable computation budget would likely not close the performance gap.
>
> Below we detail the reasons (We have added this analysis in the Section 4.2 in our revised paper).
>
> 1. **It's NOT about *More* reasoning, but a *Different Kind* of Reasoning.**
>
> A baseline agent, even with a larger budget for more steps, is stuck in a reactive "Think-Then-Act" loop. It attempts to reason about the *entire task* based on an initial, incomplete, and often flawed perception of the world (e.g., the first few lines of a large file). In this case, more computation budget just results in a more elaborate *wrong* plan.
>
> In contrast, ROGA is proactive, which employs **Active World Modeling (AWM)** mechanism (implemented as comprehension–operation separation). As detailed in the revised **Section 3.2**, For the AWM, the agent use the reasoning budget for a meta-cognitive loop, iteratively identifying knowledge gaps and dynamically generating specialized "comprehension tools" to explore the environment and piece together a coherent world model. This shifts the agent from a reactive "read-act" loop to a proactive "probe-model-act" cycle, a fundamental change in how agents interact with large-scale, partially observable worlds.
>
> 2. **Brute-Force Budgeting Cannot Fix an Algorithmic Flaw.**
>
> A baseline agent with more tokens will likely lead to a **High-Cost Failure,** which produce a longer, more confident, but still incorrect plan based on limited observation of the world. It will spend its budget on generating complex code that fails because it's operating on incorrect assumptions about the file's structure or content. Its failure is architectural, not resource-based.
>
> In contrast, ROGA spend higher token budget on Active World Modeling, which yields a much higher rate of return. The implemented comprehension phase de-risk the entire rest of the process. While the upfront cost is higher, the *overall* *cost per successful task* is far lower because it avoids the fundamental failures common in baselines. The significant jump in Pass@1 rate for a moderate increase in cost demonstrates that ROGA's framework is more computationally efficient at achieving the final goal.

---

> ### Author Response · Authors · 2025-11-21
> **Response to Question 2**
>
> > **Response to Question 2:** In the Dual-Reflection mechanism, how often does semantic validation detect issues missed by functional testing, and could examples illustrate these cases?
>
> Thank you for this interesting question. To better understand these failure modes, we manually examined 20 cases randomly selected from our experiments where SSC was triggered. Among them, 14 cases pass the functional testing (i.e., successful executed) but fail in semantic validation, triggering another generation-validation loop. These errors fall into two main categories.
>
> 1. **Failure to Propagate State (Incorrect Return Value):** This occurs when a tool performs an operation correctly but fails to return the updated object. For example, a tool is generated to modify a table object (e.g., sort a column). The code correctly performs the sort in its local scope but lacks a "return" statement for the modified table object. In this case, functional testing passes because no error is thrown during execution. However, our semantic validation flags this as an error because it recognizes that a tool intended to modify a stateful object must return that object for further use. Without the return value, the modification is lost and cannot be used by any subsequent tools.
> 2. **Ignoring the Current State (Incorrect Input Parameters):** This error happens when a generated tool is executable but fails to use the relevant, up-to-date object from **Persistent Symbolic Memory**. For example, after filtering a table in the previous step, the next tool should create a chart based on the filtered table. However, the tool is instead operating on the reopened table object. In this case, functional testing passes because no error is thrown during execution. Yet, the semantic validation detects that the tool is not using the existing variable handle.
>
> In essence, functional testing confirms a tool is *operationally valid*, while semantic validation ensures it is *contextually and logically correct* within the agent's ongoing, stateful workflow. This two-layer validation is indispensable for achieving the high reliability ROGA demonstrates on long-term, stateful tasks.

---

> ### Author Response · Authors · 2025-11-21
> **Response to Question 3**
>
> > **Response to Question 3:** The paper shows that ROGA uses significantly more tokens than baselines. What fraction of this overhead comes from comprehension versus reflection, and how does this affect cost-benefit trade-offs?
>
> Thank you for this important question. The higher token usage is a deliberate design choice, which is an upfront investment in reasoning that directly yields a significantly higher task success rate.
>
> To address this question, we manually inspected 10 randomly selected cases in the OSWorld benchmark for ROGA. We find that the overhead primarily stems from two parts:
>
> 1. **Active World Modeling (Comprehension, ~40% of overhead):** This is the cost of building an accurate world model from a partially observable environment. Unlike baselines that fail by acting on incomplete information, ROGA iteratively probes the environment to understand it. To clearly guide the agent through this **Comprehension** process, we must provide the LLM with detailed prompts (causing more input tokens). This upfront cost is essential for success in complex tasks; our ablation study shows removing it causes an 11.82-point drop in task success.
> 2. **Dynamic Capability Evolution (Reflection, ~20% of overhead):** This is the cost of validating and refining tools to ensure they are reliable and reusable.
>
>     On one hand, **Situated Self-Correction** requires providing the LLM with the context from the Persistent Symbolic Memory (PSM), which increases token consumption. However, this produces more reliable tools and thus a higher success rate. Our ablation study shows removing this component causes an 8.18-point drop in task success.
>
>     Conversely, the **Capability Lifecycle** actually *mitigates* long-term costs. Our ablation shows that removing it leads to a **1.54x increase in reasoning steps** due to redundant tool regeneration. This investment prevents the much higher downstream costs of task failures and repeated generation.
>
>
> In summary, prevailing methods are fundamentally unable to solve long-term, stateful, and partially observable tasks, no matter the token budget. The resulting task failures incur high human labor costs, which are far more expensive than tokens. ROGA, as a new paradigm, significantly boosts the success rate within a reasonable token budget, while actively minimizing long-term costs by reusing validated capabilities.

---

> ### Author Response · Authors · 2025-11-21
> **Response to Question 4**
>
> > **Response to Question 4:** What are the main failure cases (e.g., comprehension errors, tool misuse, sandbox inconsistencies), and do they suggest directions for further refinement?
>
> Thank you for this excellent question and it is also questioned by **Reviewer 8iBf (Question 1)**. After an in-depth analysis on the failures, we find that for long-term, stateful tasks, the error types, from most to least critical, are **(1)** **execution errors** caused by a loss of state continuity (sandbox inconsistencies in your question), **(2) comprehension failures** due to partial observability, and **(3)** **Tool generation errors**.
>
> We detail the analysis below, supported by our ablation study (Table 2) in the revised paper (We have also added this analysis in the Section 4.2 and Appendix C in our revised paper).
>
> 1. **Stateful Execution Failures / Sandbox Inconsistencies (The Most Critical Error Class):** The single largest performance drop in our ablation study (**-12.73 points** in Pass@1) occurred when we removed **Persistent Symbolic Memory (PSM)**. This quantitatively demonstrates that the inability to maintain state across multiple, dependent actions is the primary failure mode. Prevailing agents, operating on a stateless model, frequently corrupt their working state or fail on iterative tasks (e.g., modifying a data table in step 2 and then creating a chart from that *modified* table in step 5). Their tool calls are isolated, breaking the chain of causality and leading to execution errors because the evolved state of the world has been lost. ROGA's PSM, with its atomicity and rollback mechanisms, directly and successfully mitigates this class of errors.
> 2. **Comprehension Failures:** The second-largest performance drop (**-11.82 points**) occurred when we removed **Active World Modeling (AWM)**. This shows that failing to build an accurate model of a partially observable environment is another major source of error. When an agent cannot see the entire context (e.g., a large spreadsheet), it acts on incomplete or incorrect assumptions. This leads to fundamental flaws in planning and tool specification, causing the agent to pursue the wrong goal entirely. For instance, it might attempt to operate on non-existent data or only on the partially-observed range of cells in Excel.
> 3.  **Tool Generation Failures:** Removing **Situated Self-Correction** led to a significant but smaller performance drop (**-8.18 points**). This indicates that generating faulty tools is a frequent but less fundamental source of error. An agent might still succeed if a task requires a simple tool. However, without a state-aware validation process, the likelihood of generating syntactically incorrect or semantically misaligned tools (a form of tool misuse) increases substantially, leading to incorrect results.
>
> **Further Refinement Directions:** Further refinement could centers on extending ROGA's principles from ensuring internal consistency to managing external interaction and higher-level strategy. For instance, Active World Modeling could evolve into an interactive disambiguation system, enabling the agent to proactively ask users for clarification when it detects ambiguity. Furthermore, our self-correction mechanism could be elevated from the tool level to perform plan-level validation, allowing the agent to simulate and debug an entire multi-step strategy before execution. Moreover, the robustness of our Persistent Symbolic Memory could be hardened against external system failures through advanced checkpointing, allowing the agent to recover and resume long-horizon tasks after a crash.

---

> ### Author Response · Authors · 2025-11-21
> **Response to Question 5**
>
> > **Response to Question 5:** Given that the improvements on TableBench and SheetCopilotBench are modest and the gains do not extend to Math500, how generalizable is ROGA beyond the office domain?
>
> Thank you for your question. We respectfully disagree with the premise that the improvements are modest or that the results show a lack of generalizability. In fact, experiments in section 4.2 and 4.3 show that ROGA outperform all the generalist agent baselines and even surpass the highly specialized agent. Additionally, comparable results on Math500 are not evidence of a limitation, but rather direct generalization confirmation of our central thesis: **ROGA introduces a new paradigm designed to solve a specific class of problems,** long-term, stateful tasks in partially observable environments**, where previous agent paradigms fail, without sacrificing** their performance on established tasks. We have revised our paper in Section 4.2 to make this point clearer.
>
> Below we detail our response in two parts.
>
> 1. **Outperforming a Specialist Is a Sign of a Powerful General Paradigm.** The gains on TableBench and SheetCopilotBench are conceptually significant. It is crucial to note that ROGA, a **general-purpose framework**, not only matches but **outperforms** a highly-specialized, SOTA agent (SheetAgent) in its own domain. This is a powerful validation of our approach, demonstrating that ROGA's foundational principles are robust enough to surpass domain-specific engineering.
> 2. **Comparable Performance on Math Confirms Generality, Not Limits It.** The fact that ROGA performs on par with other agents on Math500 is an expected and crucial result. Math problems are typically stateless and fully observable, meaning they do not require ROGA's core innovations (e.g., Active World Modeling, Persistent Symbolic Memory). Therefore, achieving comparable performance demonstrates that our framework successfully **extends** agent capabilities to a new, harder class of problems **without sacrificing** existing ones. We have built a more capable generalist agent, not a narrower, specialized one.
>
> Besides, in our revised paper, we have added additional cross-domain studies to demonstrate its generalizability. We utilize **MMLU-Pro**, a challenging multi-task understanding benchmark, to test a different and critical capability of generalization: knowledge-intensive reasoning. As shown in our revised paper (Table 5, also shown below), demonstrates strong and competitive performance across both domains. The results further indicate that ROGA's design successfully enables it to tackle complex, stateful tasks while maintaining performance on a broad spectrum of general-purpose reasoning benchmarks.
>
> | Method | Math500 (Exec@1) | Math500 (Pass@1) | MMLU-Pro (Exec@1) | MMLU-Pro (Pass@1) |
> | --- | --- | --- | --- | --- |
> | AutoAgent | 84.00 | 54.00 | 100.00 | 78.00 |
> | OctoTools | 100.00 | 70.00 | 97.00 | 64.00 |
> | OWL | 99.00 | 71.00 | 93.00 | 65.00 |
> | ROGA | 99.00 | 70.00 | 100.00 | 76.00 |

---

> ### Author Response · Authors · 2025-11-27
> **Request for Reviewer's Feedback**
>
> Dear Reviewer jaLz,
>
> We are very grateful for your thorough review and the important points you raised.
>
> We wanted to follow up on our earlier response to see if it has satisfactorily addressed your concerns. Your perspective has been crucial for strengthening our work's core contributions. We remain available and eager to discuss any points you might still have.
>
> We hope that our revisions and clarifications have earned your support for our paper.
>
> Sincerely,
>
> Authors

---

### Official Review · Reviewer_8iBf · 2025-11-01

**Soundness:** 2
**Presentation:** 2
**Contribution:** 2
**Rating:** 4
**Confidence:** 3

**Summary:**

This paper presents ROGA, a framework for automatic tool generation (ATG) agents designed specifically for office productivity tasks involving Excel, Word, and PowerPoint. The authors identify three key limitations of current ATG agents: (1) insufficient handling of long file contexts, (2) lack of context sharing across tool calls, and (3) inefficient tool reuse. ROGA addresses these through four innovations: a comprehension-operation paradigm, dual-reflection tool generation, state-sharing sandbox execution, and finite-state machine-based tool lifecycle management. Experiments show ROGA outperforms existing ATG agents by up to 13.64% on office benchmarks.

**Strengths:**

ROGA consistently outperforms baselines across multiple benchmarks (OSWorld, WAA, GAIA-Office, TableBench, SheetCopilotBench) and even matches specialized agents. Table 2 systematically demonstrates the contribution of each component, with state-sharing showing the most significant impact.

**Weaknesses:**

The paper focuses extensively on tool generation but lacks discussion of recent work on tool retrieval and approaches that combine generation with retrieval.
The main contribution appears to be thoughtful engineering rather than fundamental algorithmic innovation. The paper would benefit from more clearly articulating what is novel versus what adapts existing techniques.

**Questions:**

I have some questions in the systematic failure analysis:
What types of errors occur most frequently? (comprehension failures, tool generation errors, execution errors?)
When does dual-reflection fail to catch errors?
Are there systematic biases in which file types or task categories cause problems?

---

> ### Author Response · Authors · 2025-11-21
>
> Thank you for your valuable feedback and insightful review. We are delighted that you recognized the empirical strengths of our work, particularly ROGA's consistent outperformance across multiple benchmarks and the systematic demonstration of each component's contribution. Your constructive critique has been instrumental in helping us improve our paper. We will now address your specific concerns in detail, followed by detailed answers to your questions.

---

> ### Author Response · Authors · 2025-11-21
> **Rebuttal to Weakness 1: Lack of discussion on tool retrieval.**
>
> > Rebuttal to Weakness 1: Lack of discussion on tool retrieval.
>
> Thank you for your valuable feedback and for highlighting the need to better situate our work within the context of tool retrieval literature.
>
> Based on the discussed recent work on retrieval-based tool generation in the original manuscript, we have thoroughly revised our manuscript to extend the discussion to make it sufficient, primarily by restructuring and expanding the Related Work section (Section 2.1).
>
> Specifically, our revisions address your comments in the following ways:
>
> 1. **Discussion of Tool Retrieval:** We have added a new dedicated paragraph on "Tool Retrieval" in Section 2.1. This section now discusses seminal works like Toolformer, ToolLLM, and ToolGen. We clarify that while these methods are powerful for selecting and invoking tools from a *predefined* set, they are limited in open-ended environments where the necessary tools do not yet exist. This distinction helps to better motivate the need for the automatic tool generation (ATG) paradigm that our work advances.
> 2. **Inclusion of Hybrid Retrieval-Generation Approaches:** We have also updated the "Automatic Tool Generation" paragraph to include recent approaches that combine retrieval with generation. We now cite and discuss methods that leverage code retrieval and prompt-based retrieval to improve the quality of generated tools. This acknowledges the synergy between the two fields.
> 3. **Clarifying the Novelty of ROGA:** Most importantly, incorporating these related works allows us to more sharply define ROGA's unique contribution. We now explicitly argue that both tool retrieval and existing ATG methods often operate on the assumption of a well-defined task or a clear tool specification. In contrast, ROGA is designed for complex, partially observable environments (like office tasks) where the primary challenge is to first *understand the environment and infer the tool's requirements* through what we term "Active World Modeling." Our framework focuses on the preceding, crucial step of reasoning over long, ambiguous contexts to derive the tool specifications, a problem not directly addressed by the works you mentioned. This positions ROGA as a complementary and necessary advancement for applying agentic capabilities to long-horizon, stateful tasks.
>
> We believe these revisions, as seen in the updated Section 2.1, now provide a more complete and accurate picture of the research landscape and better highlight the specific contributions of ROGA. Thank you again for your constructive suggestion.

---

> ### Author Response · Authors · 2025-11-21
> **Rebuttal to Weakness 2: Contribution is "thoughtful engineering" rather than "fundamental algorithmic innovation".**
>
> > **Rebuttal to Weakness 2: Contribution is "thoughtful engineering" rather than "fundamental algorithmic innovation".**
>
> We sincerely thank you for raising the important point about the distinction between "thoughtful engineering" and "fundamental algorithmic innovation." We agree that this distinction is crucial, and we appreciate the opportunity to clarify the core novelty of our work.
>
> We respectfully argue that our contributions represent a fundamental paradigm shift for agents in long-term, stateful environments. While the components we designed are indeed carefully engineered, they are not a collection of disparate tricks. Instead, they are the concrete instantiations of a new, coherent agent paradigm designed to overcome the specific, fundamental limitations of prevailing reactive models.
>
> To make this explicit, we have **significantly revised our paper, especially Section 3, to reframe our contributions around three core algorithmic innovations**, moving from a component-level description to a principled, paradigm-level one.
>
> 1. **Active World Modeling (AWM):**
>     - **Limitation Addressed:** Prevailing agent paradigms assume full observability, where the agent passively perceives a complete state. This fails in realistic office tasks where contexts (e.g., a large spreadsheet) are too vast to fit in a context window, making them partially observable.
>     - **Our Fundamental Innovation (AWM):** We introduce **Active World Modeling**, a paradigm where the agent does not just passively read but *actively probes* the environment to build its own world model. As detailed in the revised **Section 3.2**, the agent enters a meta-cognitive loop, iteratively identifying knowledge gaps and dynamically generating specialized "comprehension tools" to explore the environment and piece together a coherent model. This shifts the agent from a reactive "read-act" loop to a proactive "probe-model-act" cycle, a fundamental change in how agents interact with large-scale, partially observable worlds.
> 2. **Persistent Symbolic Memory (PSM):**
>     - **Limitation Addressed:** Prevailing paradigms are built on a memory-less execution model, where tool calls are stateless and isolated. This breaks the chain of causality needed for iterative tasks (e.g., modifying the same file object across multiple steps).
>     - **Our Fundamental Innovation (PSM):** We propose **Persistent Symbolic Memory**, a formal mechanism that provides the agent with a structured, symbolic state ledger to track the evolution of objects and states over time. As described in the revised **Section 3.3**, this is not just a shared memory space; it is an explicit state ledger that enables temporal reasoning and stateful interaction by maintaining symbolic handles to objects across action steps. Our "State-Sharing Sandbox" is the engineered system that enforces the principles of PSM, including atomicity and reversibility, to guarantee the integrity of this state chain. This is a fundamental departure from stateless execution models.
> 3. **Dynamic Capability Evolution (DCE):**
>     - **Limitation Addressed:** Prevailing paradigms rely on static capability generation, where tools are created one-shot for immediate needs. This leads to redundant regeneration and prevents the agent from learning from past experience.
>     - **Our Fundamental Innovation (DCE):** We introduce **Dynamic Capability Evolution**, which reframes tool generation as a continuous meta-learning process on the agent’s own capabilities. As explained in the revised **Section 3.4**, this paradigm is realized through two integrated mechanisms: **a formal Finite-State Machine** to manage the lifecycle of capabilities (retaining, refining, and reusing them) and a **Situated Self-Correction process** to drive their evolution based on contextual feedback. This transforms the agent from a simple tool-maker into a self-improving system that adapts its skills over time—a fundamental step towards long-term agent autonomy.
>
> In summary, we have **revised the manuscript (Abstract, Introduction, and Section 3)** to clearly articulate that ROGA's components are implementations of these three fundamental concepts—AWM, PSM, and DCE. Together, they form a new paradigm that directly addresses the core failures of existing reactive agents in long-term, stateful environments. We believe these revisions clarify that our work offers not just thoughtful engineering, but a principled and foundational contribution to the design of generalist agents.
>
> Thank you again for your valuable feedback, which has helped us significantly improve the clarity of our paper.

---

> ### Author Response · Authors · 2025-11-21
> **Response to Question 1: What types of errors occur most frequently?**
>
> Thank you for this insightful question. In the revised paper, we have conducted an in-depth systematic analysis on different error types, which demonstrate the effectiveness of ROGA design. Our analysis, strongly supported by the ablation study in our paper (Table 2), indicates that for long-term, stateful tasks, the error types for prevailing agents, from most to least critical, are **(1)** execution errors caused by a loss of state continuity, **(2)** comprehension failures due to partial observability, and **(3)** Tool generation errors.
>
> We detail the analysis below (We have also added this analysis in the Section 4.2 and Appendix C in our revised paper).
>
> 1. Stateful Execution Failures are the most critical error class: The single largest performance drop in our ablation study (-12.73 points in Pass@1) occurred when we removed Persistent Symbolic Memory (PSM). This quantitatively demonstrates that the inability to maintain state across multiple, dependent actions is the primary failure mode for agents in this domain. Prevailing agents, operating on a stateless execution model, frequently fail at iterative tasks (e.g., modifying a data table in step 2 and then creating a chart from that modified table in step 5). Their tool calls are isolated, breaking the chain of causality and leading to execution errors or incorrect final outputs because the evolved state of the world has been lost. ROGA's PSM directly and successfully mitigates this class of errors.
> 2. Comprehension Failures represent the second-largest performance drop (-11.82 points), occurred when we removed Active World Modeling (AWM). This shows that failing to build an accurate model of the partially observable environment is another major source of error. When an agent cannot see the entire context (e.g., a large spreadsheet), it acts on incomplete or incorrect assumptions. This leads to fundamental flaws in planning and tool specification, causing the agent to pursue the wrong goal entirely. For instance, it might attempt to operate on non-existent data, or only on the partially-observed range of cells in Excel.
> 3. Tool Generation Failures: Removing Situated Self-Correction led to a significant, but smaller, performance drop (-8.18 points). This indicates that generating faulty tools is a frequent, but less fundamental, source of error compared to the two issues above. An agent might still succeed if a task requires a simple tool or if it gets lucky on a regeneration attempt. However, without a state-aware validation process, the likelihood of generating syntactically incorrect or semantically misaligned tools increases substantially, leading to incorrect results.

---

> ### Author Response · Authors · 2025-11-21
> **Response to Question 2: When does dual-reflection fail to catch errors?**
>
> This is an excellent question. While our **Situated Self-Correction (SSC)** mechanism (implemented as dual-reflection) is highly effective, it is not infallible. Its primary strength lies in catching functional errors (e.g., syntax errors, incorrect API usage) and contextual inconsistencies within the agent's current state. However, it can fail to catch more subtle, higher-level logical errors.
>
> To better understand these failure modes, we manually examined 20 cases randomly selected from our experiments where SSC was triggered, but the task ultimately failed to pass the benchmark's check. Our analysis reveals that the failures typically fall into two main categories (We have also added this analysis in the Appendix C in our revised paper):
>
> 1. **Nuance Semantic Flaws** is the most common failure mode. It occurs when the agent generates a tool that is functionally correct, but it performs the wrong semantic action due to the ambiguity in the user's request. For example, a user request "create a chart based on the data in this table." The agent correctly generates a tool to create the chart. However, the request does not specify *where* to place the chart. The agent makes a reasonable decision to place it on a new sheet. While this action is valid and successfully executed, the benchmark's ground truth might expect the chart to be placed on the *current* sheet. Finally, the task fails due to this semantic mismatch with the unstated expectation.
> 2. **Subtle Variable Confusion** can occur during long-term tasks where the **Persistent Symbolic Memory** becomes populated with numerous variables and objects. If several objects have similar names, types, or descriptions (e.g., `df_filtered`, `df_final`, `df_summary`), the agent might occasionally select the wrong one for an operation. This error is particularly insidious because the generated tool can be both syntactically correct and semantically reasonable for the incorrectly chosen variable.
>
> We believe these limitations are not flaws in the SSC mechanism itself, but rather inherent challenges of natural language ambiguity and complex state management in open-ended tasks. They represent an important and exciting direction for future research, potentially involving interactive clarification with the user or more advanced state representation techniques. Thank you for pushing us to clarify these boundaries.

---

> ### Author Response · Authors · 2025-11-21
> **Response to Question 3: Are there systematic biases in which file types or task categories cause problems?**
>
> Thank you for this excellent question. To address it, we conduct a detailed breakdown of results on the OSWorld benchmark. The results show a clear performance difference across file types. The performance differences arise because different tasks, to varying degrees, highlight the limitations of the current agent paradigm in long-term, stateful, and partially observable environments.The detailed analysis not only confirms this trend but also reveals that as task complexity increases, the performance gap between ROGA and existing paradigms widens, powerfully validating our approach.
>
> Here is a detailed breakdown of our findings. We have also added this analysis in the Appendix C in our revised paper.
>
> | File Type | Agent | Exec@1 (%) | Pass@1 (%) |
> | --- | --- | --- | --- |
> | **Word** | **ROGA** | **100.00** | **42.86** |
> |  | AutoAgent | 66.67 | 28.57 |
> |  | OctoTools | 80.95 | 14.29 |
> |  | OWL | 90.48 | 4.76 |
> | **Excel** | **ROGA** | **91.49** | **31.91** |
> |  | AutoAgent | 51.06 | 10.64 |
> |  | OctoTools | 46.81 | 14.89 |
> |  | OWL | 72.34 | 17.02 |
> | **PPT** | **ROGA** | **97.62** | **26.19** |
> |  | AutoAgent | 73.81 | 19.05 |
> |  | OctoTools | 73.81 | 23.81 |
> |  | OWL | 80.95 | 21.43 |
>
> This detailed comparison yields several key insights:
>
> 1. **Consistent Superiority Across All File Types:** ROGA consistently and significantly outperforms all baseline agents across every file type, in terms of both metrics. This demonstrates the general robustness of our paradigm.
> 2. **Performance Scales with Complexity:**
>
>     **Baseline agents exhibit erratic and unpredictable performance.** For instance, OctoTools and OWL achieve their lowest success rates on Word tasks, which are structurally the simplest, yet perform better on more complex PPTX files. AutoAgent, conversely, struggles most with the Excel. This unpredictable performance suggests the current paradigms are brittle and lack a systematic approach to handling different forms of complexity.
>
>     **In stark contrast, ROGA’s performance scales predictably with the inherent complexity of the file types.** Its success rate is highest on structurally simpler Word files (42.86%), followed by state-intensive Excel files (31.91%), and then the most complex multi-modal PPTX files (26.19%). This graceful degradation demonstrates that ROGA's paradigm provides a robust and principled way to manage increasing task complexity. While baselines fail unpredictably, ROGA's performance reflects a systematic and more effective approach to the underlying challenges.
>
> 3. **ROGA Excels in Execution Reliability:** A critical finding is ROGA's near-perfect execution rate (100% on Word, 91.5% on Excel, 97.6% on PPTX). In contrast, baselines frequently fail to even generate executable code, especially on complex Excel tasks (e.g., OctoTools' 46.8% execution rate). This shows that ROGA's framework, particularly its **Situated Self-Correction**, reliably produces syntactically correct and contextually valid tools. While baselines fail at a fundamental level, ROGA successfully overcomes the execution challenge, allowing it to focus on the higher-level problem of semantic correctness.
>
> In summary, the performance variations are not arbitrary. The erratic behavior of baselines highlights the brittleness of existing paradigms. In contrast, ROGA's consistent superiority and predictable scaling with complexity provide strong evidence that our paradigm (**Active World Modeling, Persistent Symbolic Memory, and Dynamic Capability Evolution)** is correctly identifying and solving the fundamental bottlenecks in long-term, stateful environments.

---

> ### Author Response · Authors · 2025-11-27
> **Request for Reviewer's Feedback**
>
> Dear Reviewer 8iBf,
>
> Thank you again for your detailed and constructive review.
>
> We are writing to gently follow up on our rebuttal and to ensure our responses have effectively addressed your concerns. Your insights have been instrumental in helping us refine the paper, and we are ready and willing to provide any further clarification you might need.
>
> We appreciate your time and look forward to your updated assessment.
>
> Sincerely,
>
> Authors

---

### Author Response · Authors · 2025-11-21

Dear Reviewers, ACs:

We sincerely thank all reviewers for their thoughtful and constructive feedback on our paper. We are especially grateful for the recognition of our work's **key strengths**, including the systematic identification of challenges in generalist agents (Reviewers SvUL, jaLz, akoz), the clear motivation study (Reviewers jaLz, SvUL), the well-structured writing (Reviewers SvUL, jaLz, akoz), the strong empirical results that show consistent outperformance (Reviewers 8iBf, jaLz, SvUL, akoz) and good generalization (Reviewers jaLz, SvUL), and significant practical impact (Reviewer jaLz).

We also deeply appreciate the insightful suggestions for improvement. We are particularly grateful for the feedback (from **Reviewer jaLz** and **8iBf**) that pushed us to move beyond a component-level description and articulate the core conceptual novelty of our work. Inspired by this, we have undertaken a **major revision** of the manuscript to reframe our contributions around the fundamental **paradigm shift** that ROGA represents.

To clarify this shift and better highlight our algorithmic innovations, we have mapped our original engineering-focused terminology to the new paradigm-level concepts that they instantiate. As suggested by **Reviewer jaLz**, this reframing clarifies that ROGA’s components are not just "thoughtful engineering" but concrete implementations of a principled and foundational contribution to agent design.

| **Original Engineering Terminology (in Original Paper)** | **Revised Paradigm-Level Concept (in Revised Paper)** | **Core Limitation Addressed** |
| --- | --- | --- |
| Comprehension–Operation Paradigm | **Active World Modeling (AWM)** | Failure to build a coherent world model from long, partially observable contexts. |
| State-Sharing Sandbox | **Persistent Symbolic Memory (PSM)** | Memory-less execution model that fails to track state evolution in iterative tasks. |
| Dual-Reflection + Tool Lifecycle FSM | **Dynamic Capability Evolution (including Situated Self-Correction & A Formalism for Capability Lifecycle)** | Static, one-shot capability generation that leads to redundant regeneration. |

Based on all the valuable feedback, we have made the following major revisions to the paper:

1. **Reframed Core Novelty Around a New Agent Paradigm (Reviewers jaLz, 8iBf, akoz):** We have significantly revised the Abstract, Introduction, and Methodology (Section 3) to articulate ROGA's three core algorithmic innovations: **Active World Modeling (AWM)**, **Persistent Symbolic Memory (PSM)**, and **Dynamic Capability Evolution (DCE)**. This reframing clarifies that our work introduces a new, coherent paradigm for agents in **long-term, stateful, partially observable environments**, particularly evidenced in Office tasks, rather than just a collection of engineered components.
2. **Expanded Discussion on Related Work (Reviewer 8iBf):** We have restructured and expanded the Related Work section (Section 2.1) to include a dedicated discussion on **Tool Retrieval** (e.g., Toolformer, ToolLLM) and hybrid retrieval-generation approaches. This helps to more sharply define ROGA's unique contribution in addressing the preceding challenge of inferring tool requirements from partially observable environments.
3. **Strengthened Cross-Domain Generalization Study (Reviewers akoz, jaLz, SvUL):** To further demonstrate ROGA's generalizability beyond office tasks, we have expanded our evaluation to include the **MMLU-Pro** benchmark, a challenging test of knowledge-intensive reasoning. The results (Table 5) show that ROGA maintains strong, competitive performance, confirming that our paradigm extends agent capabilities without sacrificing performance on general tasks.
4. **Clarified Cost-Benefit Analysis (Reviewers jaLz, akoz):** We extended the "Cost Analysis" in Section 4.2 to explain that ROGA’s higher token usage is a strategic investment in a more robust reasoning paradigm (AWM and DCE) that prevents costly task failures. We also highlight that our **Dynamic Capability Evolution** mechanism serves as a built-in cost mitigation strategy by amortizing tool generation costs over time.
5. **Added In-Depth Qualitative and Quantitative Analysis (Reviewers 8iBf, jaLz, SvUL) in Appendix C:** We added the detailed analyses to provide deeper insight into why our system works, including:
    - A systematic **failure case analysis** that quantitatively links our ablation results to the primary error types.
    - An analysis of the **limitations of our self-correction mechanism**.
    - A **performance breakdown by file type**.
6. **Provided Implementation Details (Reviewer akoz) in** **Appendix D.**

We believe these extensive revisions have significantly strengthened the paper and more clearly articulated its core scientific contributions. We thank all reviewers once again for their invaluable guidance, which has been instrumental in improving our work.

Sincerely,
Authors

---

### Meta-Review · Area_Chair_pmQ2 · 2026-01-12

**Summary:**

The reviews can be clustered into the following points:

1. Novelty: multiple reviewers (jaLz, 8iBf) thought the original framing read as “thoughtful engineering”, and asked for a clearer statement of what is fundamentally new vs. a clean integration of known patterns.

2. Literature: 8iBf asked for a more complete and up-to-date related works with respect to tool retrieval and retrieval-augmented generation approaches.

3. Efficiency/scalability: jaLz and akoz raised concern that the framework’s cost is materially higher ( in terms of token or steps) and that Pass-1 remains moderate on OSWorld-style tasks, thus the cost–benefit case needed to better quantified.

4. Analysis: Several reviewers requested more insights into why it works (for examples, which errors are reduced, when reflection fails, which task categories remain hard) and how the components map to gains.

**Reviewer Concerns:**

Addressed by the rebuttal / revision
- Related work on tool retrieval (8iBf).
The revised related work explicitly discusses tool retrieval (e.g., ToolLLM/ToolGen) and distinguishes them as relying on predefined toolsets, then situates ATG and hybrid retrieval+generation methods  while arguing that these typically assume well-defined tool specifications.
- “Just more steps” and different reasoning paradigm (jaLz, 8iBf). The revision reframes the core contributions as AWM, emphasizing iterative probing under partial observability rather than a single passive read step. It also formalizes and describes SSC as a state-grounded dual-channel validator.
- Ablation study.
The ablation table shows large Pass@1 drops when removing AWM and PSM, consistent across OSWorld/WAA/GAIA-Office.
- Cost discussion is more explicit (jaLz, akoz).
- Cross-domain “non-office” checks expanded (SvUL, akoz, jaLz).
The revised paper includes Math500 and MMLU-Pro results, showing performance broadly comparable to strong baselines rather than degrading.
- Domain-specific agent comparison for spreadsheets (SvUL).

Still (partially) outstanding
- The concerns of the novelty still remains
- The efficiency claims rely on limited sampling or coarse attribution
- The Reproducibility still depends on artifact release.

**Reviewer Scores:**

- Reviewer 8iBf (4 to 5 likely): Their two main weaknesses—(i) missing tool retrieval context and (ii) novelty articulation—are directly addressed
- Reviewer jaLz (4 to 4.5/5): jaLz’s core hesitation is conceptual novelty at a top venue plus cost.
- Reviewer SvUL (4 to 5 likely): The rebuttal addresses (i) domain motivation and (ii) “is it office-specific?” reasoning
- Reviewer akoz (4 to 5 likely): The major ask was reproducibility details and broader generalization. The revised text includes SSC mechanism description and non-office eval results

I would expect the panel to move from 4 on average to something like three 5s and one 4/5

---

### Decision · Program_Chairs · 2026-01-26

Accept (Poster)